# Next-Token Prediction and Regret Minimization

## Abstract

We consider the question of how to employ next-token prediction algorithms in adversarial online decision making environments. Specifically, if we train a next-token prediction model on a distribution $\mathcal{D}$ over sequences of opponent actions, when is it the case that the induced online decision making algorithm (by approximately best responding to the model's predictions) has low adversarial regret (i.e., when is $\mathcal{D}$ a *low-regret distribution*)?

For unbounded context windows (where the prediction made by the model can depend on all the actions taken by the adversary thus far), we show that although not every distribution $\mathcal{D}$ is a low-regret distribution, every distribution $\mathcal{D}$ is exponentially close (in TV distance) to one low-regret distribution, and hence sublinear regret can always be achieved at negligible cost to the accuracy of the original next-token prediction model. In contrast to this, for bounded context windows (where the prediction made by the model can depend only on the past $w$ actions taken by the adversary, as may be the case in modern transformer architectures), we show that there are some distributions $\mathcal{D}$ of opponent play that are $\Theta(1)$-far from any low-regret distribution $\mathcal{D}'$ (even when $w = \Omega(T)$ and such distributions exist). Finally, we complement these results by showing that the unbounded context robustification procedure can be implemented by layers of a standard transformer architecture, and provide empirical evidence that transformer models can be efficiently trained to represent these new low-regret distributions.

## 1 Introduction

Large language models are trained to perform well at the task of next-token prediction: given some substring of text, estimate the conditional distribution of the next word/token. Increasingly, there is a focus on using these models to perform a far broader set of tasks, including making strategic decisions on our behalf (Chen et al., 2021; Park et al., 2025; Krishnamurthy et al., 2024; Nie et al., 2025).

Consider the problem of training such a model to play a repeated game (e.g., repeated rock-paper-scissors). Like in next-token prediction, the model has to take the actions taken in the game so far (a subsequence of tokens) and, from this, come up with a new mixed action to take (a distribution over next tokens). If we think of the tokens as the adversary's actions, then it even makes sense that playing well in this game directly corresponds to how well our model can predict the next token. Where things differ is in how these tokens are generated – instead of being stochastically sampled from a large data set, they are adversarially chosen by an opposing player who wants the model to fail. One basic property we might desire from these models in such settings is *adversarial regret minimization*. That is, regardless of what actions the adversary takes, our model does at least as well as if it always played the best fixed action in hindsight.

This raises the question: are regret minimization and next-token prediction compatible goals? When is it the case that training a next-token predictor on a dataset (e.g., of game transcripts) will produce a low-regret learning algorithm? Are there ways to automatically augment a data set with more data so the resulting models have less regret? What alternatives to next-token prediction are there when training these models?

## 1.1 OUR RESULTS

We study an online decision making setting where a decision maker needs to take actions in response to a changing state of nature (e.g. an adversary's action in a game, a current stock price, etc.). The decision maker has access to a next-token prediction model trained on some distribution of state sequences, and would like use the predictions from this model to help them make utility optimizing decisions.

Importantly, they would like to perform well not just when the true distribution of states is drawn from the distribution their model is trained on but also when the sequence of states is controlled by an adversary. This leads to the question of whether it is possible to *robustify* a decision-model: take a model $\mathcal{M}_0$ and produce a model $\mathcal{M}$ that represents a similar distribution over sequences as $\mathcal{M}_0$, while guaranteeing low regret against any adversary.

We prove the following results.

- First, we remark that there exist next-token prediction models such that if a decision maker approximately best responds to these predictions (e.g., via a quantal best response), they guarantee sublinear regret. In particular, quantal best responses to the Polya urn process closely simulate the classical Hedge learning algorithm (**Theorem 2.3**).

- Second, we positively answer the question of robustification by showing that given any next-token prediction model $\mathcal{M}_0$ it is possible to produce a model $\mathcal{M}$ such that i. quantal best responses to the predictions of $\mathcal{M}$ lead to sublinear regret, and ii. the TV distance between the distributions represented by $\mathcal{M}$ and $\mathcal{M}_0$ is arbitrarily small (**Theorem 3.1**).

- We then shift our attention to prediction models with bounded context length (i.e., prediction models whose outputs can only depend on the previous $L$ tokens). In contrast to the previous result, we show that such models are in general impossible to robustify (**Theorem 4.1**). However, if the robustified model is allowed to use a larger context length $L'$, it is possible to produce a robust model with $O(1/\sqrt{L' - L})$ per-round regret (**Theorem 4.2**).

- Finally, we address the question of whether it is actually possible to *train* robust models, with a focus on transformer models. We provide two pieces of evidence towards an affirmative answer to this question. First, we show that transformer models can effectively represent the robustified models of Theorem 3.1 with a mild increase in size (**Theorem 5.1**). Second, we provide experimental evidence that it is possible to train small transformers to represent robustified versions of simple distributions (**Section 5.2**).

## 1.2 RELATED WORK

We discuss additional related work in more detail in Appendix A.

## 2 MODEL AND PRELIMINARIES

**Notation** We use $\mathbb{I}[A]$ to denote the indicator function of expression $A$, which takes the value 1 when $A$ is true, and 0 otherwise. We generally denote sequences of elements in bolded letters (e.g., $\boldsymbol{\theta}$), elements of these sequences with subscripts ($\theta_t$), and subsegments of these sequences with superscripts ($\boldsymbol{\theta}^{a:b} = (\theta_a, \theta_{a+1}, \ldots, \theta_b)$, $\boldsymbol{\theta}^b = (\theta_1, \ldots, \theta_b)$). Full proofs are generally deferred to Appendix B for the sake of brevity.

## 2.1 NEXT-TOKEN PREDICTION

The problem of *next-token prediction* can be formally stated as follows. We are given a distribution $D \in \Delta(\Theta^T)$ over sequences of $T$ tokens from an alphabet $\Theta$. The goal is to learn a (next-token prediction) *model* $\mathcal{M}$ that, given as input any prefix token sequence $\boldsymbol{\theta}^{t-1} = (\theta_1, \ldots, \theta_{t-1})$, outputs the conditional distribution of the next token given this prefix, which we denote by $\mathcal{M}(\boldsymbol{\theta}^{t-1}) \in \Delta(\Theta)$. We write $\mathcal{M}(\theta \mid \boldsymbol{\theta}^{t-1})$ to denote the probability of a specific token $\theta \in \Theta$ in the distribution $\mathcal{M}(\boldsymbol{\theta}^{t-1})$.

By iterating the operation of next token prediction, any candidate solution $\mathcal{M}$ to the next token prediction problem induces its own distribution $D(\mathcal{M})$ over sequences of $T$ tokens. In particular,

we can define

$$\Pr_{\boldsymbol{\theta}^T \sim D(\mathcal{M})}[\boldsymbol{\theta}^T] = \mathcal{M}(\theta_1|\emptyset)\mathcal{M}(\theta_2|\theta_1)\mathcal{M}(\theta_3|\theta_1, \theta_2) \cdots \mathcal{M}(\theta_T|\boldsymbol{\theta}^{T-1}).$$

Conversely, every distribution $D$ corresponds[1] to some model (in the sense that it is induced by a collection of conditional distribution functions $\mathcal{M}(\theta_t|\boldsymbol{\theta}^{t-1})$). We can therefore measure the quality of a solution $\mathcal{M}$ to the next-token prediction problem via the TV distance $d_{TV}(D, D(\mathcal{M}))$ between the true distribution and the distribution induced by the model. Likewise, we can measure the similarity between two models $\mathcal{M}$ and $\mathcal{M}'$ via the TV distance of their respective distributions.

**Bounded context length** Later in the paper, we will consider models that have the additional restriction of *bounded context length* – that is, the model's prediction $\mathcal{M}(\theta_t|\boldsymbol{\theta}^{t-1})$ for the $t$th token can only depend on the $w$ preceding tokens $(\theta_{t-w}, \theta_{t-w+1}, \ldots, \theta_{t-1})$ for some window size $w$. We defer further discussion of bounded context lengths to the beginning of Section 4.

## 2.2 Adversarial Online Decision Making

The second problem we consider is that of (adversarial) *online decision making*. In this problem, a *decision maker* interacts with an *adversary* over the course of $T$ rounds. In each round $t \in [T]$ of interaction, the learner takes an action (specifically, a mixed action $\pi_t \in \Delta(A)$ supported on some finite action set $A$) while, simultaneously, an adversary selects a state $\theta_t \in \Theta$. As a result of this interaction, the decision maker receives expected utility $\mathbb{E}_{a_t \sim \pi_t}[U(a, \theta_t)]$, where the utility function $U : A \times \Theta \to [-1, 1]$ is known to all parties and fixed over time (we extend $U$ linearly to mixed strategies of the decision maker by writing $U(\pi, \theta) = \mathbb{E}_{a \sim \pi}[U(a, \theta)]$). After this interaction, the state $\theta_t$ chosen by the adversary is revealed to the learner, who can then use this information in the selection of their subsequent actions.

The goal of the decision maker is to maximize their cumulative utility over all $T$ rounds. Of course, the extent to which they can do so depends on the adversarial choices of $\theta_t$ taken by the adversary (notably, unlike in the next-token prediction problem, the sequence of states $\boldsymbol{\theta}^T = (\theta_1, \theta_2, \ldots, \theta_T)$ is not necessarily sampled from some distribution $D$). Despite this, one of the fundamental results in the theory of online learning shows that regardless of the actions taken by the adversary, it is possible for the decision maker to obtain sublinear *regret*: the gap between their cumulative utility and the cumulative utility of the best fixed action in hindsight. Formally, given a sequence of (mixed) actions $\boldsymbol{\pi} = (\pi_1, \ldots, \pi_T)$ and states $\boldsymbol{\theta} = (\theta_1, \ldots, \theta_T)$, we define the external regret as

$$\text{ExtReg}(\boldsymbol{\pi}, \boldsymbol{\theta}) = \max_{a^* \in A} \frac{1}{T} \sum_t [U(a^*, \theta_t) - U(\pi_t, \theta_t)].$$

One algorithm that guarantees sublinear regret for the decision maker is the Hedge algorithm (Freund & Schapire, 1997). The Hedge algorithm chooses $\pi_t$ so that (for any $a \in A$) $\pi_t(a) \propto \exp\left(\frac{1}{\sqrt{T}} \sum_{s=1}^{t-1} U(a, \theta_s)\right)$. It can be shown that this guarantees that $\text{ExtReg}(\boldsymbol{\pi}, \boldsymbol{\theta}) = O(\sqrt{(\log|A|)/T})$, regardless of the sequence of states chosen by the adversary.

## 2.3 Interplay between Next-Token Prediction and Regret Minimization

One natural way to apply a next-token prediction algorithm to the problem of online decision making is by using it to predict the sequence of adversary states. In particular, the decision maker can use an algorithm for next-token prediction to predict the next state, and then play the optimal action conditioned on this state. Formally, for any distribution $\mu \in \Delta(\Theta)$ over states, let $\text{BR}(\mu) = \arg\max_{a \in A} \mathbb{E}_{\theta \sim \mu}[U(a, \theta)]$ be the decision maker's *best response* action to this distribution. In online decision making in *stochastic settings* (where the sequence of states $\boldsymbol{\theta}$ is drawn from some distribution $D$), best responding to the predictions of an accurate model leads to zero external regret.

**Lemma 2.1.** *Let $D \in \Delta(\Theta^T)$ be a distribution over sequences of $T$ states, and let $\mathcal{M}$ be a next-token prediction model that has perfectly learned the distribution $D$ ($D(\mathcal{M}) = D$). Consider the algorithm for the decision maker which sets $\pi_t = BR(\mathcal{M}(\boldsymbol{\theta}^{(t-1)}))$ (that is, the best response to the model's prediction of state at time $t$). Then the expected regret of the decision maker on sequences sampled from $D$ is at most zero, i.e., $\mathbb{E}_{\boldsymbol{\theta} \sim D}[\text{ExtReg}(\boldsymbol{\pi}, \boldsymbol{\theta})] \leq 0$.*

---

[1] For mathematical convenience, we will assume that all distributions $D$ we consider have full support – that is, every sequence in $\Theta^T$ appears with some positive (albeit possibly arbitrarily small) probability in the distribution. Under this assumption, this correspondence is bijective.

However, we would like stronger guarantees than this – ideally, we would like to construct an online decision making algorithm with *adversarial* regret guarantees (e.g., those obtained by Hedge). This leads to the question: does there exist a distribution $D$ where the online decision making algorithm constructed in Lemma 2.1 incurs $o(1)$ regret against any adversary? Unfortunately, the answer to this question is negative, as the following lemma demonstrates.

**Lemma 2.2.** *Let $\mathcal{M}$ be a next-token prediction model. There exists a utility function $U$ such that, if the decision maker sets $\pi_t = BR(\mathcal{M}(\boldsymbol{\theta}^{(t-1)}))$, there exists an adversarial sequence of states $\boldsymbol{\theta} \in \Theta^T$ that induces high regret, i.e., with the property that $\mathrm{ExtReg}(\boldsymbol{\pi}, \boldsymbol{\theta}) = \Omega(1)$.*

Ultimately, the negative result in Lemma 2.2 follows from the fact that the learning algorithms constructed by best responding to a sequence of next-token prediction are *deterministic* (in the sense of always playing pure actions in $A$).

We can attempt to sidestep this issue by introducing noise in the best response of the decision maker. One natural and well-studied way to do this is to replace the best response with a *quantal best response*[2]. Given a distribution $\mu \in \Delta(\Theta)$ over states and a parameter $\eta > 0$, we define the quantal best response $\mathrm{QBR}(\mu, \eta) \in \Delta(A)$ to be the mixed action that plays action $a \in A$ with probability proportional to $\exp(\frac{1}{\eta} U(a, \mu))$. Note that as $\eta \to 0$, this approaches the deterministic best response (and as $\eta \to \infty$, this approaches the uniform distribution over all actions).

We define the *Polya urn model* $\mathcal{M}_{\text{Polya}}$ to be the following next-token prediction model: for any $t \in [T]$, we let

$$\mathcal{M}_{\text{Polya}}(\theta | \boldsymbol{\theta}^{(t-1)}) = \frac{1 + \sum_{s=1}^{t-1} \mathbb{I}\left[\theta_s = \theta\right]}{|\Theta| + (t-1)}. \tag{1}$$

Intuitively, the probability of seeing a specific token $\theta$ at round $t$ is roughly equal to the empirical probability of observing $\theta$ in the string so far. More accurately, it is exactly the fraction of tokens equal to $\theta$ in the string $\mathrm{Str}(\Theta) + \boldsymbol{\theta}^{(t-1)}$, where $\mathrm{Str}(\Theta)$ is an arbitrary concatenation of all the tokens in $\Theta$ (it is necessary to add this additional term so that equation 1 is well-defined for $t = 1$, and so that the induced distribution $D(\mathcal{M}_{\text{Polya}})$ has full support). The following lemma shows that quantal best responses to predictions of the Polya urn model guarantee adversarial low regret.

**Lemma 2.3.** *Consider the algorithm for the decision maker which sets $\pi_t = QBR(\mathcal{M}_{Polya}(\boldsymbol{\theta}^{(t-1)}), \eta)$, for $\eta = 1/\sqrt{T}$. Then for any adversarial sequence of states $\boldsymbol{\theta} \in \Theta^T$,*

$$\mathrm{ExtReg}(\boldsymbol{\pi}, \boldsymbol{\theta}) = O\left(\frac{\log T + \log |A|}{\sqrt{T}}\right).$$

Motivated by Lemma 2.3, we say that a next-token model $\mathcal{M}$ is a *low-regret model* if quantal best responses to this model guarantee $o(1)$ worst-case regret; formally, for any adversarial sequence of states $\boldsymbol{\theta} \in \Theta^T$, the sequence of mixed actions $\boldsymbol{\pi} \in \Delta(A)^T$ defined via $\pi_t = \mathrm{QBR}(\mathcal{M}(\boldsymbol{\theta}^{t-1}), 1/\sqrt{T})$ satisfies $\mathrm{ExtReg}(\boldsymbol{\pi}, \boldsymbol{\theta}) = o(1)$.

**Example (Adversarial Online Prediction)** By selecting the utility function $U$ appropriately, the online decision making framework can be made to capture a wide range of different possible applications. One particularly relevant example (that we will use as a running example throughout the remainder of this paper) is the problem of *adversarial online prediction*.

In this problem, we set the action set $A$ equal to the state space $\Theta$, and define $U(a, \theta) = \mathbb{I}\left[a = \theta\right]$; that is, the decision maker receives a point if they successfully predict the current state (and receives zero points otherwise). In some later applications (e.g., the experiments in Section 5.2), we will further insist that actions and states are binary ($A = \Theta = \{0, 1\}$).

Note that in this example, the goals of the online decision maker and the next-token prediction algorithm are very closely aligned – they both want to produce good predictions of the next state, but with slightly different metrics of success (adversarial regret guarantees versus statistical distance guarantees). One consequence of this is that we can directly interpret the quantal best response as sampling from the next-token prediction model with temperature $\eta$.

---

[2]This response function is also known under many other names, including *softmax response*, *Boltzmann exploration*, and *multinomial logit response*.

## 3 ROBUSTIFICATION WITH UNBOUNDED CONTEXT LENGTH

Lemma 2.3 demonstrates that the Polya urn model is a low-regret model – following its recommendations (by quantally best responding to them) will result in adversarial low-regret guarantees for an online decision maker. While it is possible to construct other low-regret models similarly, not every model is low-regret. For example, the model $\mathcal{M}$ for binary states ($\Theta = \{0, 1\}$) which always predicts the next bit to be 1 with probability $1/3$ can be shown to incur $\Omega(1)$ regret against adversarial sequences of states (e.g., if the adversary selects the all-zero sequence of states $\theta_t = 0$, this model will never predict the next state correctly).

This raises a natural question. Assume we have access to a next-token model $\mathcal{M}_0$. Can we "robustify" our model and obtain a new model $\mathcal{M}$ that is both low-regret and close to the original model $\mathcal{M}_0$ (in the sense that the distributions $D_0$ and $D$ they induce are similar in TV distance)?

In this section, we answer this question affirmatively. In Algorithm 1, we give a procedure for taking an arbitrary next-token prediction model $\mathcal{M}_0$ and transforming it into a low-regret next-token prediction model $\mathcal{M}$. The key idea is to only modify the behavior of the model on prefixes $\boldsymbol{\theta}^t$ where the model has already incurred high regret (by arguments similar to those in Lemma 2.1, this should happen with low probability if the sequence of states truly is sampled from $D(\mathcal{M}_0)$). On such high-regret prefixes, we instead draw the prediction of the model from a Polya urn model, guaranteeing low-regret on the remainder of the time horizon.

---

**Algorithm 1** Robustification of a next-token prediction model

---

**Require:** Next-token prediction model $\mathcal{M}_0$ implementing distribution $D_0$, sequence of states $\boldsymbol{\theta}^{t-1}$, utility function $U : A \times \Theta \to [-1, 1]$, parameter $\alpha > 0$.
**Ensure:** Outputs $\mathcal{M}(\boldsymbol{\theta}^{t-1})$ for some model $\mathcal{M}$ implementing a low-regret distribution $D$.
    **for** $s = 1 \ldots t - 1$ **do**
        Define $\pi_s \leftarrow \text{QBR}(\mathcal{M}_0(\boldsymbol{\theta}^{s-1}), 1/\sqrt{T})$ *(the mixed action of a quantal best response to the original model)*.
        Define $\pi_{\text{HEDGE},s} \leftarrow \text{QBR}(\mathcal{M}_{\text{Polya}}(\boldsymbol{\theta}^{s-1}), \frac{1}{\sqrt{T}})$ *(the mixed action of a quantal best response to Polya urn model)*
        Define $\text{REGRET}_s \leftarrow \text{EXTREG}(\boldsymbol{\pi}^s, \boldsymbol{\theta}^s)$
        Define $\text{REGRET}_{\text{HEDGE},s} \leftarrow \text{EXTREG}(\boldsymbol{\pi}^s_{\text{HEDGE}}, \boldsymbol{\theta}^s)$
        **if** $\text{REGRET}_s \geq \text{REGRET}_{\text{HEDGE},s} + \frac{1}{\sqrt{T}} \log |A| + \sqrt{8(1 + \alpha)(\log T)/s}$ **then**
                                  $\triangleright$ *(We are out-of-distribution, return prediction of Polya urn model)*
            **return** $\mathcal{M}_{\text{Polya}}(\boldsymbol{\theta}^{t-1})$
        **end if**
    **end for**
                                $\triangleright$ *(We are in distribution, return original model prediction)*
    **return** $\mathcal{M}_0(\boldsymbol{\theta}^{t-1})$

---

**Theorem 3.1.** *Running Algorithm 1 on a model $\mathcal{M}_0$ (with $D_0 := D(\mathcal{M}_0)$) results in a robustified model $\mathcal{M}$ (with $D := D(\mathcal{M})$) with the following properties:*

- *$\mathcal{M}$ is a low-regret model with worst-case regret $O\left(\frac{1}{\sqrt{T}} \log(|A| \cdot T) + \sqrt{(1 + \alpha) \log T}\right)$.*

- *The TV distance between $D$ and $D_0$ is bounded by $d_{TV}(D, D_0) \leq |A| T^{-\alpha}$.*

## 4 ROBUSTIFICATION WITH A BOUNDED CONTEXT LENGTH

In the previous section, we concerned ourselves with next-token prediction models whose prediction of the state $\theta_t$ at time $t$ could depend on all previous states $\boldsymbol{\theta}^{t-1}$. In practice, most next-token prediction models (e.g. those based on transformer architectures) are autoregressive models restricted by a context length $L$. That is to say, the model's prediction $\mathcal{M}(\theta_t | \boldsymbol{\theta}^{t-1})$ is a round-independent function of the previous $L$ tokens $\boldsymbol{\theta}^{(t-L):(t-1)} = (\theta_{t-L}, \theta_{t-L+1}, \ldots, \theta_{t-1})$. When $t \leq L$, then $\mathcal{M}(\theta_t | \boldsymbol{\theta}^{t-1})$ can be an arbitrary function of the past tokens (as in the unbounded context case). We will refer to such models as *L-bounded models* for short.

As before, every bounded context model $\mathcal{M}$ induces a distribution $D(\mathcal{M})$ over state sequences of length $T$, and as before, we will measure the similarity of two models by the TV distance of their induced distributions.

We still would like to use these models to aid in adversarial online decision making[3]. Of course, the limited context window of these models constrains what regret guarantees are possible. The setting of *online learning with bounded recall* studies online decision making instances where the action at round $t$ must be a function of the previous $L$ losses (i.e., states). It can be shown (Schneider & Vodrahalli, 2024) that in this setting, there are simple modifications of Hedge that guarantee at most $O(L^{-1/2})$ regret against any adversary, and that this regret bound is tight (intuitively, this regret bound is achievable by restarting Hedge every $L$ rounds).

As in the unbounded context setting, we can use an $L$-bounded model $\mathcal{M}$ to solve online learning with $L$-bounded recall by playing quantal best responses to the predictions of $\mathcal{M}$. In particular, we can show (in analogy to Lemma 2.3) that there exist $L$-bounded models $\mathcal{M}$ where if the decision maker plays $\pi_t = \text{QBR}(\mathcal{M}(\boldsymbol{\theta}^{t-1}), 1/\sqrt{L})$, the decision maker guarantees $O(1/\sqrt{L})$ regret for themselves.

We are then faced with the same question as in the previous section: if we start with an existing $L$-bounded next-token prediction model $\mathcal{M}_0$, can we robustify it into a model $\mathcal{M}$ that is similar to $\mathcal{M}_0$ but also obtains optimal worst-case regret guarantees against an adversary?

### 4.1 IMPOSSIBILITY WITH THE SAME CONTEXT LENGTH

We begin by demonstrating that, unlike in the unbounded context setting, robustification of bounded context models is in general impossible, even in very simple online decision-making settings (e.g. the adversarial online prediction problem with binary states).

Intuitively, this is because there can exist different $L$-bounded models $\mathcal{M}_0$ and $\mathcal{M}_1$ that induce very different distributions $D(\mathcal{M}_0)$ and $D(\mathcal{M}_1)$ over sequences of length $T$ (in particular, almost never agreeing about the next token), but that share the same distribution of substrings of length $L$. In particular, an $L$-bounded model that can only ever see substrings of length $L$ will have trouble distinguishing whether the state sequence is being generated by $\mathcal{M}_0$ or $\mathcal{M}_1$. If the goal is to robustify $\mathcal{M}_0$, $\mathcal{M}$ then has the impossible tradeoff between playing predictions close to that of $\mathcal{M}_0$ (guaranteeing low TV distance, but possibly incurring high regret with respect to sequences drawn from $\mathcal{M}_1$) or playing predictions that guarantee low regret for $\mathcal{M}_1$ (which cause a large TV distance with respect to $\mathcal{M}_0$).

**Theorem 4.1.** *Set $L = T/2$, $A = \Theta = \{0, 1\}$, and $U(a, \theta) = \mathbb{I}[a = \theta]$ (the binary adversarial online prediction task). There exists a context length $L$ model $\mathcal{M}_0$ (with $D_0 = D(\mathcal{M}_0)$) such that for any other context length $L$ model $\mathcal{M}$ (with $D = D(\mathcal{M})$), either:*

1. *The TV distance $d_{TV}(D_0, D) > 1/24$ (i.e., the two models are not close).*

2. *There exists an adversarial sequence of states $\boldsymbol{\theta} \in \Theta^T$ such that if $\boldsymbol{\pi} \in \Delta(A)^T$ is the sequence of quantal best responses to $\mathcal{M}$ ($\pi_t = QBR(\mathcal{M}(\boldsymbol{\theta}^{t-1}), 1/\sqrt{L})$), then $\text{EXTREG}(\boldsymbol{\pi}, \boldsymbol{\theta}) > 1/24$. (That is, the model $\mathcal{M}$ is not a low-regret model).*

### 4.2 ROBUSTIFICATION WITH A LONGER CONTEXT LENGTH

In the previous section, we showed that there is no way to robustify an existing $L$-bounded model $\mathcal{M}_0$ to a low-regret $L$-bounded model $\mathcal{M}$ (while implementing approximately the same distribution). In this section, we show that if we allow the robustified model to have a slightly larger context window $L'$, we *can* effectively perform this robustification. Said another way, this fact implies that it is possible to learn a model that will length-generalize from the distribution of a sufficiently short sequence while maintaining no-regret guarantees in the bounded context setting (a more realistic setting for transformer-based models).

We do this by adapting the "AverageRestartHedge" algorithm of Schneider & Vodrahalli (2024), which achieves $O\left(\frac{1}{\sqrt{m}}\right)$ external regret in adversarial online learning settings with $m$-bounded

---

[3]For technical reasons, in this section we will restrict ourselves to binary action settings ($|A| = 2$). This has the consequence that the quantal best response function $\text{QBR}(\cdot, 1/\sqrt{L})$ has a convex image, which will be important for implementing some of the algorithms for online learning with bounded recall (e.g., see the second-to-last line of Algorithm 2).

recall. At a high level, this algorithm is configured with some non-constrained low-regret sub-algorithm (canonically, Hedge) as a subroutine. It then outputs the average prediction of this sub-algorithm on a uniformly randomly chosen suffix of the previous $L$ losses.

We will run a variant of this algorithm with Algorithm 1 in place of Hedge. Specifically, given an expanded context of length $L'$, we use $L$ out of $L'$ tokens are used for next-token prediction under the original distribution $D_0$. The remaining $\Delta = L' - L$ tokens can then be viewed as the actual context length given to the online algorithm in Schneider & Vodrahalli (2024). Our Algorithm 2 calls Algorithm 1 as a subroutine, which achieves an external regret of $\tilde{O}\left(\frac{1}{\sqrt{\Delta}}\right)$ (as implied by Theorem 4.1, it is impossible to get non-trival guarantees when $\Delta = 0$).

---

**Algorithm 2** Robustifying Bounded Context Models with Longer Context Lengths

---

**Require:** An existing $L$-bounded next-token prediction model $\mathcal{M}_0$, parameter $\alpha > 0$, input $\boldsymbol{\theta}^{L'}$.
**Ensure:** A robustified $L'$-bounded next-token prediction model $\mathcal{M}$ (with $L' > L$, $\Delta = L' - L$).
 Run Algorithm 1 on $\mathcal{M}_0$ with time horizon $\Delta$ to produce a robustified model $\mathcal{M}_\Delta$.
 **for** $m = L+1, \ldots, L'$ **do**
  $\mu_m \leftarrow \mathcal{M}_\Delta(\boldsymbol{\theta}^{m:L'})$ (i.e., the output of $\mathcal{M}_\Delta$ on the sequence $\theta_m, \theta_{m+1}, \ldots, \theta_L$)
 **end for**
 Choose a $\mu \in \Delta(\Theta)$ so that $\text{QBR}(\mu, 1/\sqrt{\Delta}) = \frac{1}{\Delta}\sum_{m=L+1}^{L'} \text{QBR}(\mu_m, 1/\sqrt{\Delta})$.
 **return** $\mathcal{M}(\boldsymbol{\theta}^{L'}) = \mu$.

---

**Theorem 4.2.** *Fix $L' > L$ and let $\Delta = L' - L$. Running Algorithm 2 on an $L$-bounded model $\mathcal{M}_0$ (with $D_0 := D(\mathcal{M}_0)$) results in a robustified $L'$-bounded model $\mathcal{M}$ (with $D := D(\mathcal{M})$) with the following properties:*

- *The model $\mathcal{M}$ is a low-regret model, with worst-case regret $\left(1 + \frac{\Delta}{T}\right)\left[\frac{\sqrt{2}+1}{\Delta} + \sqrt{\frac{8\log T + 8(\alpha+1)\log\Delta}{\Delta}}\right]$.*

- *The TV distance between $D$ and $D_0$ is bounded by $d_{TV}(D, D_0) \leq \Delta^{-\alpha}$.*

## 5 TRAINING LOW-REGRET TRANSFORMER MODELS

On one hand, Theorem 3.1 demonstrates that it is information theoretically possible to robustify *any* next-token prediction model $\mathcal{M}$ with negligible changes to the underlying distribution. At the same time, this raises questions about whether we can actually *train* low-regret models (after all, if $d_{\text{TV}}(D, D_0)$ is exponentially small, no training procedure can efficiently distinguish between samples drawn from $D$ and samples drawn from $D_0$

In this section we investigate this question for the special case of *transformer models*, providing evidence that it is possible to directly robustify low-regret transformer models. In Section 5.1, we show it is possible to implement the operations of Algorithm 1 in the logic of a standard transformer model (i.e., if $\mathcal{M}_0$ can be represented by a small transformer, so can $\mathcal{M}$). In Section 5.2, we provide experimental evidence showing that a simple masking procedure allows us to practically train low-regret transformer models.

### 5.1 REPRESENTING ROBUSTIFIED MODELS

In this section, we show that the representational limitations of transformers pose no obstacle to robustification. To that end, we construct a transformer that robustly predicts future states by adding a constant number of layers to a transformer that solves next-token prediction.

**Theorem 5.1.** *Suppose there exists a transformer $\mathcal{M}_0$ with $L$ layers and embedding dimension $m$ that exactly solves the next token prediction task over distribution $D_0$; that is, $\mathcal{M}_0(\theta_t|\boldsymbol{\theta}^{t-1}) = \text{Pr}_{D_0}[\theta_t|\boldsymbol{\theta}^{t-1}]$). Then, there exists a transformer $\mathcal{M}'$ with $L' = L + 4$ layers and embedding dimension $m' = m + O(1)$ that approximates the output of Algorithm 1.*

We state the theorem rigorously and present its proof in Appendix C. At a high-level, the argument relies on constructing four layers that use the outputs of $\mathcal{M}_0$ to simulate Algorithm 1. Self-attention

plays an essential role in the construction. Identifying the distribution induced by the Polya Urn strategy and calculating the two regret quantities involve computing aggregations over sequences of tokens, which are naturally simulated with self-attention layers. Our construction reflects a realistic class of transformers by maintaining tight bounds on embedding dimension and depth and employing multi-layer perceptrons that can be compactly represented as shallow ReLU networks.

## 5.2 Empirically Robustifying Simple Transformers

In the previous sections, we demonstrated the existence of a procedure for learning specialized low-regret online learning algorithms by carefully perturbing the original statistical training data. In this section, we also demonstrate that in simple settings, it is also practically efficient to train small transformers with this algorithm, suggesting that robustification procedures may be practically plausible for modifying LLM behavior for decision-making while retaining good statistical performance.

We consider the special case of a decision problem to match the state. Both the action space and the state space are binary $A = \Theta = \{0, 1\}$, and the utility function $U(a, \theta) = \mathbb{I}[a = \theta]$. We conduct experiments with the minimal decoder-only transformer, NanoDO (Liu et al., 2024). The transformer predicts a binary sequence. We adopt the default parameters of NanoDO, with a context length of $T = 1024$, 256 embedding dimensions, 4 attention heads, 3 transformer block layers, and 1024 inner dimensions.

We train the transformer on three datasets with a batch size of 128. The three training processes all converge and stop after 500 steps.

**BERNOULLI** is the in-distribution and non-robust transformer. The dataset is generated from the distribution where the first half of 512 bits are from $\mathrm{Ber}(1/3)$ and the second half are from $\mathrm{Ber}(2/3)$.

**POLYAURN** is the robust transformer without distributional information. The transformer is trained on Polya Urn sequences, where the next bit is generated from the empirical distribution in history: $\Pr[\theta_{t+1} = 1 | \theta_1, \ldots, \theta_t] = \frac{\sum_{i \in [t]} \theta_i}{t}$. By setting the temperature to $\frac{1}{\sqrt{T}}$, POLYAURN plays the same strategy as the Hedge algorithm.

**ROBUST_BERNOULLI** is trained on the robustified distribution of BERNOULLI. We do this in the following way. We sample training data from the same distribution as BERNOULLI. We also sample an equal number of Polya Urn sequences. For a Polya Urn sequence, we keep it only if transformer BERNOULLI has a regret higher than $\frac{\alpha}{\sqrt{t}}$ for some $t \leq T$, with $\alpha = 1.5$. In other cases, we discard the sequence. To keep the TV-distance unchanged in the training process, we mask out the loss calculation over the prefix of a Polya urn sequence, up to the first position $t$ where there is a regret higher than $\frac{\alpha}{\sqrt{t}}$. By masking out the prefix, the transformer does not learn the distribution that generates a high-regret prefix.

### 5.2.1 Regret Evaluation

We evaluate the regret of the three transformers on eight ground truth distributions over sequences. The bits are drawn independently from each other. The first four are static distributions where each bit is drawn from either $\mathrm{Ber}(1/3)$ or $\mathrm{Ber}(2/3)$. In the other four simulations, we adopt the same simulation setup as in Schneider & Vodrahalli (2024). The bits are generated from a periodically drifting distribution with $\Pr[\theta_t = 1] = \left| \sin(\pi/6 + t \cdot \pi/\phi) \right|$, for period $\phi \in \{\frac{T}{2}, \frac{T}{5}, \frac{T}{10}, \frac{T}{20}\}$. We evaluate the regret of quantal best-response by applying a soft-max layer and setting the temperature to $\frac{1}{\sqrt{T}}$. We estimate from 128 independent sequences sampled from the ground truth distribution.

We plot the regret of the three transformers in Figure 1. The following observations validate that NanoDO learns the dataset constructed by Algorithm 1. First, The transformers effectively learn to play the robust strategy. ROBUST_BERNOULLI and POLYAURN both have vanishing regret on all eight ground truth data-generating processes. Second, ROBUST_BERNOULLI learns the switching policy of Algorithm 1. ROBUST_BERNOULLI preserves the same in-distribution regret of BERNOULLI in plot $(1, 2)$, which is negative around $-0.16$.

### 5.2.2 TV-distance

We report the estimated TV-distance between the models in Table 1. We estimate from 128 independent sample of sequences and report the $95\%$ confidence interval. We also test the TV-distance

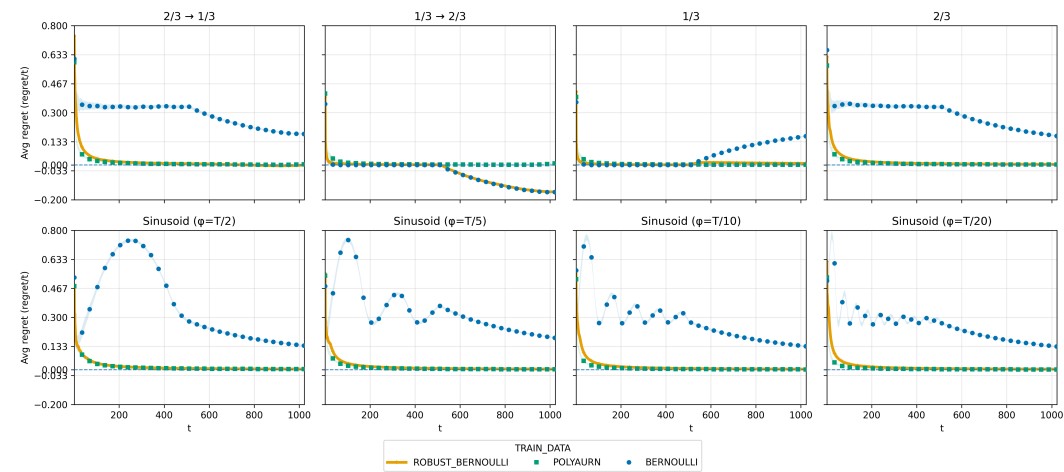

Figure 1: The regret of three transformers over 8 ground truth distributions. The three transformers are 1) ROBUST_BERNOULLI, robustified BERNOULLI, 2) POLYAURN, and 3) BERNOULLI. The eight ground truth distributions are: a) half Ber(2/3) and then half Ber(1/3); b) half Ber(1/3) and half Ber(2/3), the same distribution that ROBUST_BERNOULLI and BERNOULLI were trained on; c) Ber(1/3); d) Ber(2/3); and four periodically changing distributions on the second row. The plot shows (very narrow) confidence intervals in light color.

|  | BERNOULLI$_1$ | ROBUST_BERNOULLI | POLYAURN |
|---|---|---|---|
| BERNOULLI$_1$ | — | $0.7602 \pm 0.0267$ | $1.0000 \pm 0$ |
| BERNOULLI$_2$ | $0.4193 \pm 0.0232$ | $0.6869 \pm 0.0295$ | $1.0000 \pm 0$ |

Table 1: The TV-distance between transformers. BERNOULLI$_i$ are two models trained on the same Ber(1/3)→Ber(2/3) process with different random seeds.

between two models trained on the same BERNOULLI distribution, but with different random seeds. ROBUST_BERNOULLI achieves a lower TV-distance than POLYAURN, where POLYAURN has a TV-distance estimated as high as $1.0000$ from the original distribution BERNOULLI.

In addition to the TV-distance, we report the Next-Token TV-distance here. As shown in Table 1, the full-sequence TV-distance is brutally strict and even high for two models trained on the same distribution. Tiny per-token differences are calculated as a difference across the entire sequence. Even models that behave similarly at a token level can have a high TV-distance on whole sequences. Per-step TV instead measures the local difference of the two predictive models at each prefix.

We define the following Next-Token TV-distance. For each prefix $\boldsymbol{\theta}^s$, we can calculate the TV-distance of the next-token prediction, $d_{\text{TV}}(\mathcal{M}_1(\cdot|\boldsymbol{\theta}^s), \mathcal{M}_2(\cdot|\boldsymbol{\theta}^s))$. The Next-Token TV-distance $d_{\text{NT}}$ takes the expectation of the prefix from the distribution of BERNOULLI, i.e., with the first $T/2$ drawn from Ber(1/3) and the second $T/2$ tokens from Ber(2/3): $d_{\text{NT}} = \mathbb{E}_{\boldsymbol{\theta} \sim \text{BERNOULLI}} \left[ \frac{1}{T} \sum_{s \in [T]} d_{\text{TV}}(\mathcal{M}_1(\cdot|\boldsymbol{\theta}^s), \mathcal{M}_2(\cdot|\boldsymbol{\theta}^s)) \right]$.

We report the Next-Token TV-distance in Table 2. The results are calculated with 128 independent draws of a sequence.

|  | BERNOULLI$_1$ | ROBUST_BERNOULLI | POLYAURN |
|---|---|---|---|
| BERNOULLI$_1$ | — | $0.0199 \pm 0.0001$ | $0.1529 \pm 0.0003$ |
| BERNOULLI$_2$ | $0.0156$ | $0.0299 \pm 0.0001$ | $0.1655 \pm 0.0004$ |

Table 2: Next-Token TV distance between transformers. BERNOULLI$_i$ are two models trained on the same Ber(1/3)→Ber(2/3) process with different random seeds.

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

## A ADDITIONAL RELATED WORK

Our study is at the intersection of decision-making in online learning as well as modern transformer architectures in deep learning.

Classically, there have been many studies of online decision-making for model families defined by classes of finite automata (Rubinstein, 1986; Ben-porath, 1990; Lehrer & Solan, 2009; Piccione & Rubinstein, 1993), though these earlier works are typically in the context of repeated games (which, while related, is distinct from the online learning setting we study in this work). We can view the connection to this earlier work by considering a transformer to implement a class of finite automata.

Park et al. (2025) is a particularly relevant modern study that studies the behavior of large language models (LLMs) as game theoretic agents in both online learning and game theory, and is perhaps the first study to directly examine whether a transformer-based architecture can also be a no-regret agent. This work focuses more on the empirical behavior of existing LLMs and also defines a complex regret-based training objective by which to train transformers. Comparatively, we present distinct and simpler algorithms to achieve the goal of low-regret transformer models and present results in different settings, focusing on the relation between next-token-prediction and regret.

Marsden et al. (2024) is another work investigating connections between length generalization in next-token-prediction and online sequence prediction, albeit in the specialized setting of online linear dynamical systems.

The idea of using transformers for decision-making has also long been present in the deep learning literature. Chen et al. (2021) proposed to use transformer models for decision-making, and Nie et al. (2025) has built on this work in the era of large language models, exploring multiple algorithms for transforming an existing LLM into a model that can perform in-context exploration in online decision-making settings. Krishnamurthy et al. (2024) also explores the connection between LLMs and decision-making settings, but again focuses on existing LLMs and investigating multi-arm bandit environments via online in-context learning. Finally, Vallinder & Hughes (2024) proposes another approach to modify the behavior of LLM agents in an online decision-making setting via evolving prompts.

## B OMITTED PROOFS

### B.1 PROOF OF LEMMA 2.1

*Proof.* Let $h_{t-1} := (\theta_1, \ldots, \theta_{t-1})$ denote the history up to time $t - 1$. By assumption, the model's next token prediction is the true conditional probability at every history:

$$\mathcal{M}(h_{t-1}) = D(\cdot \mid h_{t-1}).$$

At round $t$, the decision maker plays the best response to this next-token prediction:

$$\pi_t \in \mathrm{BR}\big(\mathcal{M}(h_{t-1})\big) \in \arg\max_{\pi \in \Delta(A)} \mathbb{E}_{\theta_t \sim D(\cdot \mid h_{t-1})}\big[U(\pi, \theta_t)\big].$$

For any other action $a^* \in A$. By the optimality of $\pi_t$ under the correct conditional,

$$\mathbb{E}\big[U(\pi_t, \theta_t) \,\big|\, h_{t-1}\big] \geq \mathbb{E}\big[U(a^*, \theta_t) \,\big|\, h_{t-1}\big].$$

Taking expectations over $h_{t-1}$ and using the tower property yields

$$\mathbb{E}\big[U(\pi_t, \theta_t)\big] \geq \mathbb{E}\big[U(a^*, \theta_t)\big].$$

Summing over $t = 1, \ldots, T$ gives

$$\mathbb{E}_{\boldsymbol{\theta} \sim D}\Big[\sum_{t=1}^{T} U(\pi_t, \theta_t)\Big] \geq \mathbb{E}_{\boldsymbol{\theta} \sim D}\Big[\sum_{t=1}^{T} U(a^*, \theta_t)\Big] \quad \text{for every } a^* \in A.$$

Equivalently, the expected (average) *utility regret* versus the best fixed action in hindsight is $\leq 0$:

$$\mathbb{E}_{\boldsymbol{\theta} \sim D}\left[\max_{a^* \in A} \frac{1}{T} \sum_{t=1}^{T} \Big(U(a^*, \theta_t) - U(\pi_t, \theta_t)\Big)\right] \leq 0,$$

which is precisely $\mathbb{E}_{\boldsymbol{\theta} \sim D}[\mathrm{EXTREG}(\boldsymbol{\pi}, \boldsymbol{\theta})] \leq 0$. $\qquad\square$

## B.2 PROOF OF LEMMA 2.2

*Proof.* Consider the binary token space and action space $\Theta = A = \{0, 1\}$, with utility function $U(a, \theta) = \mathbb{I}[a = \theta]$. Fix any next-token model $\mathcal{M}$ and the induced (deterministic) decision rule $\pi_t = \mathrm{BR}(\mathcal{M}(\boldsymbol{\theta}^{(t-1)}))$, which is a deterministic function of the history $h_{t-1} = (\theta_1, \ldots, \theta_{t-1})$.

Define an adversary that, after observing $a_t$ (or equivalently inferring $a_t$ from the history), sets $\theta_t = 1 - a_t$. Then the learner obtains zero utility each round:

$$U(a_t, \theta_t) = 0 \quad \text{for all } t,$$

so $\sum_{t=1}^T U(a_t, \theta_t) = 0$.

On the other hand, for the realized state sequence $\theta_{1:T}$, the best fixed action in hindsight is the majority element of the sequence, which achieves utility at least $T/2$. Therefore,

$$\mathrm{EXTREG}(\boldsymbol{\pi}, \boldsymbol{\theta}) = \frac{1}{T} \left( \max_{a \in \{0,1\}} \sum_{t=1}^T U(a, \theta_t) - \sum_{t=1}^T U(a_t, \theta_t) \right) \geq \frac{1}{2}.$$

Thus, the regret is bounded below by a constant, i.e. $\Omega(1)$. $\qquad\square$

## B.3 PROOF OF LEMMA 2.3

*Proof of Lemma 2.3.* Fix a finite action set $A$ and utilities $U : A \times \Theta \to [-1, 1]$. At round $t$, the Polya urn predictor is

$$\mathcal{M}_{\mathrm{Polya}}(\theta \mid \theta_{1:t-1}) = \frac{1 + \sum_{s=1}^{t-1} \mathbb{I}[\theta_s = \theta]}{|\Theta| + (t - 1)}.$$

The QBR with parameter $\eta > 0$ plays

$$\pi_t(a) \propto \exp\left( \tfrac{1}{\eta} U\left(a, \mathcal{M}_{\mathrm{Polya}}(\cdot \mid \theta_{1:t-1})\right) \right),$$

where

$$\exp\left( \tfrac{1}{\eta} U\left(a, \mathcal{M}_{\mathrm{Polya}}(\cdot \mid \theta_{1:t-1})\right) \right) = \exp\left( \frac{1}{\eta\left(|\Theta| + t - 1\right)} \left[ \sum_{\theta \in \Theta} U(a, \theta) + \sum_{s=1}^{t-1} U(a, \theta_s) \right] \right).$$

Thus, QBR plays

$$\pi_t(a) \propto \exp\left( \tfrac{C(a)}{\eta\left(|\Theta| + t - 1\right)} \right) \cdot \exp\left( \tfrac{1}{\eta\left(|\Theta| + t - 1\right)} \sum_{s=1}^{t-1} U(a, \theta_s) \right).$$

Thus $\pi_t$ is an *exponential-weights* distribution over actions with round-$t$ learning rate $\lambda_t := \frac{1}{\eta\left(|\Theta| + t - 1\right)}$ applied to the realized utilities $U(a, \theta_s)$, and with an action-dependent prior factor that only changes by a common (rescaling-invariant) temperature at each $t$. Standard analysis of Hedge with time-varying learning rates (apply, e.g., the potential argument round by round) gives, for any adversarial sequence $\theta_{1:T}$,

$$\sum_{t=1}^T \left( U(\pi_t, \theta_t) - U(a^\star, \theta_t) \right) \geq -\frac{\ln |A|}{\lambda_T} - \frac{1}{2} \sum_{t=1}^T \lambda_t, \quad \text{for all } a^\star \in A,$$

using $U \in [-1, 1]$. Choosing $\eta = T^{-1/2}$ (as in the statement) yields $\lambda_t = \frac{\sqrt{T}}{|\Theta| + t - 1}$, so

$$\frac{1}{T} \sum_{t=1}^T \left( U(a^\star, \theta_t) - U(\pi_t, \theta_t) \right) \leq \frac{\ln |A|}{T \lambda_T} + \frac{1}{2T} \sum_{t=1}^T \lambda_t = \frac{\ln |A|}{\sqrt{T}} + O\left( \frac{\log T}{\sqrt{T}} \right) = O\left( \frac{\log T + \ln |A|}{\sqrt{T}} \right).$$

which implies the regret bound $\mathrm{EXTREG}(\boldsymbol{\pi}, \boldsymbol{\theta}) = O\left( \frac{\log T + \ln |A|}{\sqrt{T}} \right)$. $\qquad\square$

### B.4   Proof of Theorem 3.1

*Proof.* Let $\mathcal{E}$ be the event that for some $s \in [T-1]$, $\text{REGRET}_s \geq \text{REGRET}_{\text{HEDGE},s} + \frac{1}{\sqrt{T}} \log |A| + \sqrt{8(1+\alpha)(\log T)/s}$ (i.e., we return $\mathcal{M}_{\text{Polya}}(\boldsymbol{\theta}^{t-1})$ for all rounds $t > s$). Let $\tau$ be the random variable representing the minimum such $s$; if the event $\mathcal{E}$ does not occur, let $\tau = T$.

We begin by proving that the new model $\mathcal{M}$ implements a low-regret distribution $D$. Fix any adversarial sequence of states $\boldsymbol{\theta}$ and define $\pi_t = \text{QBR}(\mathcal{M}(\boldsymbol{\theta}^{t-1}), 1/\sqrt{T})$. We can decompose the external regret $\text{EXTREG}(\boldsymbol{\pi}, \boldsymbol{\theta})$ via

$$\text{EXTREG}(\boldsymbol{\pi}, \boldsymbol{\theta}) = \max_{a^* \in A} \frac{1}{T} \left( \sum_{t=1}^{\tau} [U(a^*, \theta_t) - U(\pi_t, \theta_t)] + \sum_{t=\tau+1}^{T} [U(a^*, \theta_t) - U(\pi_t, \theta_t)] \right)$$

By assumption, for $t \leq \tau$, $M(\boldsymbol{\theta}^{t-1}) = M_0(\boldsymbol{\theta}^{t-1})$, and so $\sum_{t=1}^{\tau}[U(a^*, \theta_t) - U(\pi_t, \theta_t)] \leq \tau \text{REGRET}^{\tau}$. By the definition of $\tau$, $\text{REGRET}^{\tau} < \text{REGRET}_{\text{HEDGE},s} + \frac{1}{\sqrt{T}} \log |\Theta| + \sqrt{8(1+\alpha)(\log T)/\tau}$, and so we in turn have that $\sum_{t=1}^{\tau}[U(a^*, \theta_t) - U(\pi_t, \theta_t)] \leq \tau \left( \text{REGRET}_{\text{HEDGE},s} + \frac{1}{\sqrt{T}} \log |\Theta| + \sqrt{8(1+\alpha)(\log T)/\tau} \right) = \tau \text{REGRET}_{\text{HEDGE},s} + O\left( \sqrt{T} \left( \log |\Theta| + \sqrt{(1+\alpha) \log T} \right) \right)$.

For $t > \tau$, we have that $\mathcal{M}(\boldsymbol{\theta}^{t-1}) = \mathcal{M}_{\text{Polya}}(\boldsymbol{\theta}^{t-1})$. By Lemma 2.3, we therefore have that $\tau \text{REGRET}_{\text{HEDGE},s} + \sum_{t=\tau+1}^{T} [U(a^*, \theta_t) - U(\pi_t, \theta_t)] = O\left( T \cdot \frac{\log(T) + \log |A|}{\sqrt{T}} \right) = O(\sqrt{T} \log(|A| \cdot T))$. Combining these two terms, we have that $\text{EXTREG}(\boldsymbol{\pi}, \boldsymbol{\theta}) = O\left( \frac{1}{\sqrt{T}} (\log(|A| \cdot T) + \sqrt{(1+\alpha) \log T}) \right) = o(1)$.

We next bound the TV-distance between $D$ and $D_0$. Note that because we play the recommendation of $\mathcal{M}_0$ (and sample from $D_0$) until event $\mathcal{E}$ occurs, the TV distance $d_{\text{TV}}(D, D_0)$ is upper bounded by the probability $\Pr_{\boldsymbol{\theta} \sim D_0}[\mathcal{E}]$ of this event.

To do this, we begin by defining $\boldsymbol{a}$ to be the sequence of pure action best responses to the recommendations of $\mathcal{M}_0$; i.e., $a_t = \text{BR}(\mathcal{M}_0(\boldsymbol{\theta}^{t-1}))$. We argue that if $\boldsymbol{\theta}$ is truly sampled from $D_0$, then $\boldsymbol{a}$ and $\boldsymbol{\pi}$ obtain similar utilities and hence similar regrets. We can quantitatively bound this through the following lemma.

**Lemma B.1.** *Let $\mu \in \Delta(\Theta)$ be a distribution over states $\theta$. Let $a = BR(\mu)$ and $\pi = QBR(\mu, \eta)$. Then*

$$U(a, \mu) - U(\pi, \mu) \leq \eta \log |A|.$$

*Proof.* Note that we can equivalently define the quantal best response $\pi$ as the mixed action that maximizes the regularized utility $V(\pi) = U(\pi, \mu) + \eta H(\pi)$ (where $H$ is the entropy function). We therefore have that $V(a) \leq V(\pi)$; expanding this out (and using the fact that $H(a) = 0$), we find that $U(a, \mu) \leq U(\pi, \mu) + \eta H(\pi) \leq U(\pi, \mu) + \eta \log |A|$, from which the conclusion follows. $\square$

From Lemma B.1, it follows that when $\boldsymbol{\theta} \sim D_0$, $\mathbb{E}_{\theta_t}[U(a_t, \theta_t) - U(\pi_t, \theta_t)] \leq (\log |A|)/\sqrt{T}$. Secondly, since $a_t$ is the best response to the distribution of $\theta_t$, for any action $a_t^*$ we have that $\mathbb{E}_{\theta_t}[U(a_t^*, \theta_t) - U(a_t, \theta_t)] \leq 0$. Combining these expressions, we have that $\mathbb{E}_{\theta_t}[U(a_t^*, \theta_t) - U(\pi_t, \theta_t)] \leq (\log |A|)/\sqrt{T}$.

Let $R_t(a^*) = \sum_t (U(a^*, \theta_t) - U(\pi_t, \theta_t))$ be the unnormalized regret at time $t$ with respect to $a^*$, and similarly $R_{\text{HEDGE},t}(a^*) = \sum_t (U(a^*, \theta_t) - U(\pi_{\text{HEDGE},t}, \theta_t))$. By the previous observation, $R_t(a^*) - R_{\text{HEDGE},t}(a^*) - t(\log |A|)/\sqrt{T}$ is a super-martingale, so by Azuma's inequality, we have that

$$\Pr\left[R_t(a^*) \geq R_{\text{HEDGE},t}(a^*) + \frac{t\log|A|}{\sqrt{T}} + C\right] \leq \exp\left(-\frac{C^2}{8t}\right).$$

Substituting $C = \sqrt{8(1+\alpha)(\log T)t}$ (and normalizing by $t$), we find that

$$\Pr\left[\frac{1}{t}R_t(a^*) \geq R_{\text{HEDGE},t}(a^*) + \frac{\log|A|}{\sqrt{T}} + \sqrt{\frac{8(1+\alpha)(\log T)}{t}}\right] \leq T^{-(1+\alpha)}.$$

Now, $\text{REGRET}_t = \max_{a^*} \frac{1}{t}R_t(a^*)$. Applying a union bound over all $t \in [T]$ and $a^* \in |A|$, we have that $\Pr[\mathcal{E}] \leq |A| \cdot T^{-\alpha}$, as desired. $\qquad\square$

### B.5 PROOF OF THEOREM 4.1

To formally construct these two models $\mathcal{M}_0$ and $\mathcal{M}_1$, we will make use of the theory of *de Bruijn sequences*. The *de Bruijn graph* $\hat{G}_{\sigma,k}$ of order $k$ on an alphabet $\Theta = \{s_1, \ldots, s_{|\Theta|}\}$ of size $|\Theta|$ is a directed graph whose vertices represent all distinct sequences of length $k - 1$. For every sequence $s_{i_1}s_{i_2}\ldots s_{i_k}$ of length $k$, there is a directed edge from the vertex $s_{i_1}s_{i_2}\ldots s_{i_{k-1}}$ to the vertex $s_{i_2}s_{i_3}\ldots s_{i_k}$. A *de Bruijn sequence* of order $k$ is a cyclic sequence of characters in $\Theta$ where each of the possible $|\Theta|^k$ substrings of length $k$ appears exactly once. Note that a de Bruijn sequence of order $k$ corresponds to a (loop-removed) Eulerian cycle in a de Bruijn graph of order $k$ (which in turn must exist since every node in $\hat{G}_{\sigma,k}$ has equal indegree and outdegree $|\Theta|$).

Given a fixed de Bruijn sequence of order $L$ and over a binary alphabet $\Theta = \{0, 1\}$, we can use it to construct two nearly deterministic $L$-bounded models $\mathcal{M}_0$ and $\mathcal{M}_1$. $\mathcal{M}_0$ and $\mathcal{M}_1$ induce the same uniform marginal distribution over length $L$ substrings, but the next-token predictions are different. We construct in the following way: we define $\mathcal{M}_0(\boldsymbol{\theta}^L)$ specified by the deterministic next-token of the de Bruijn sequence, and $\mathcal{M}_1(\boldsymbol{\theta}^L) = 1 - \mathcal{M}_0(\boldsymbol{\theta}^L)$ as the deterministic opposite of $\mathcal{M}_1$. The first $L$ tokens in the Markov process are seeded uniformly so that both processes remain in the same stationary distribution: the marginal distribution over any $t > L$ substring is uniform. Thus, any context length $L$ model $\mathcal{M}$ will not be able to distinguish $\mathcal{M}_0$ from $\mathcal{M}_1$. Such a model $\mathcal{M}$ makes predictions very differently from the deterministic next-token of either $\mathcal{M}_0$ or $\mathcal{M}_1$, leading to Theorem 4.1.

*Proof.* We will prove this by constructing two $L$-bounded models $\mathcal{M}_0$ and $\mathcal{M}_1$ with the following guarantee: for any other $L$-bounded model,

$$d_{TV}(D_0, D) + \mathbb{E}_{\boldsymbol{\theta}\sim D_1}[\text{EXTREG}(\boldsymbol{\pi}, \boldsymbol{\theta})] \geq 1/12.$$

The theorem statement then follows from this guarantee (if $\mathbb{E}_{\boldsymbol{\theta}\sim D_1}[\text{EXTREG}(\boldsymbol{\pi}, \boldsymbol{\theta})] \geq 1/24$, there exists some sequence in the support of $D_1$ that realizes this).

We first describe the two models. These models will be (nearly) deterministic Markov processes of order $L$. For both $\mathcal{M}_0$ and $\mathcal{M}_1$, we set probabilities for the first $L$ tokens so that each token is equally likely to be 0 or 1 (i.e., for $t \leq w$, $\mathcal{M}_0(\theta_t|\theta^{t-1}) = \mathcal{M}_1(\theta_t|\theta^{t-1}) = \text{Unif}(\{0,1\})$).

We then use a de Bruijn sequence to set the transition probabilities of $\mathcal{M}_0$ and $\mathcal{M}_1$ as follows. Pick an arbitrary binary de Bruijn sequence of order $L$. For an $L$-tuple of states $\boldsymbol{\theta}^L = (\theta_1, \ldots, \theta_L)$, let $\text{DB}(\boldsymbol{\theta}^L) \in \{0, 1\}$ be the token immediately following $\boldsymbol{\theta}^L$ in this de Bruijn sequence. Then:

- For $\mathcal{M}_0$, set $\mathcal{M}_0(\theta_t|\boldsymbol{\theta}^{(t-L):(t-1)}) = \mathbb{I}\left[\theta_t = \text{DB}(\boldsymbol{\theta}^{(t-L):(t-1)})\right]$.

- For $\mathcal{M}_1$, set $\mathcal{M}_1(\theta_t|\boldsymbol{\theta}^{(t-L):(t-1)}) = \mathbb{I}\left[\theta_t = 1 - \text{DB}(\boldsymbol{\theta}^{(t-L):(t-1)})\right]$.

That is, we deterministically[4] set $\mathcal{M}_0$ to generate the next bit by following the de Bruijn sequence, and set $\mathcal{M}_1$ to generate the next bit by deterministically following the opposite of the de Bruijn sequence.

We begin by making the following observation: for both Markov processes $\mathcal{M}_0$ and $\mathcal{M}_1$, the uniform distribution over $L$-bit strings is a stationary distribution for the process. This follows because the induced Markov chain over $L$-bit strings is doubly stochastic for both $\mathcal{M}_0$ and $\mathcal{M}_1$ (for any state $\boldsymbol{\theta}^L$, there are exactly two predecessor states that can lead to it, one from the de Bruijn sequence, and one not from it). This means that for all $t > L$, the distribution of $\boldsymbol{\theta}^{(t-L):(t-1)}$ is uniform over $\Theta^L$.

Now let us consider the candidate robust $L$-bounded model $\mathcal{M}$ (with distribution $D(\mathcal{M}) = D$). We will call an $L$-tuple of states $\boldsymbol{\theta}^L \in \Theta^L$ *high-regret* if $\mathcal{M}(\text{DB}(\boldsymbol{\theta}^L)|\boldsymbol{\theta}^L) \geq 2/3$, and let $\alpha \in [0,1]$ equal the fraction of tuples in $\Theta^L$ that are high-regret. Note that on a high-regret sequence the prediction of $\mathcal{M}$ disagrees with that of $\mathcal{M}_1$, and will cause $\mathcal{M}$ to incur external regret on sequences drawn from $D_1$.

In particular, we first claim that $\mathbb{E}_{\boldsymbol{\theta} \sim D_1}[\text{ExtReg}(\boldsymbol{\pi}, \boldsymbol{\theta})] \geq \frac{1}{3}\alpha - (1 - \alpha)$. To see this, we will compare the expected utility of following the baseline strategy $\pi^* = (1/2, 1/2)$ to the utility of following the sequence of recommendations $\pi_t = \text{QBR}(\mathcal{M}(\boldsymbol{\theta}^{t-1}), 1/\sqrt{L})$. The baseline strategy has the property that $U(\pi^*, \theta) = 1/2$ regardless of $\theta$, so the cumulative utility of the baseline is always $T/2$. If $\boldsymbol{\theta}^{(t-L):(t-1)}$ is a high-regret tuple, then $\pi_t$ will equal $\text{DB}(\boldsymbol{\theta}^{(t-L):(t-1)})$ with probability at least $2/3$ (this probability only gets amplified by the quantal best response), and therefore $U(\pi_t, \theta_t) \leq 1/3$. On the other hand, if $\boldsymbol{\theta}^{(t-L):(t-1)}$ is not a high-regret tuple, then we only have the trivial bound $U(\pi_t, \theta_t) \leq 1$. Finally, for any $t < L$, $\theta_t$ will be drawn uniformly from $\{0,1\}$, so the expected utility $\mathbb{E}[U(\pi_t, \theta_t)] = 1/2$.

Combining these facts (and using the fact that for each $t > L$, $\boldsymbol{\theta}^{(t-L):(t-1)}$ is drawn uniformly from $\Theta^L$ and therefore has an $\alpha$ probability of being high-regret), we find that

$$\mathbb{E}_{\boldsymbol{\theta} \sim D_1}[\text{ExtReg}(\boldsymbol{\pi}, \boldsymbol{\theta})] \geq \mathbb{E}_{\boldsymbol{\theta} \sim D_1}\left[\frac{1}{T}\sum_{t=1}^{T} U(\pi^*, \theta_t) - U(\pi_t, \theta_t)\right]$$
$$\geq \frac{(T-L)}{T} \cdot \left(\frac{1}{2} - \frac{\alpha}{3} - (1-\alpha)\right)$$
$$= \frac{\alpha}{3} - \frac{1}{4}.$$

On the other hand, we will show that if $\alpha$ is too small, then the TV distance between $D_0$ and $D$ is necessarily large. Indeed, let $\tilde{D}_0$ and $\tilde{D}$ be the distributions of the first $L+1$ states from $D_0$ and $D$ respectively – by the data-processing inequality, $d_{\text{TV}}(D_0, D) \geq d_{\text{TV}}(\tilde{D}_0, \tilde{D})$. But we can directly bound $d_{\text{TV}}(\tilde{D}_0, \tilde{D}) \geq (1-\alpha)/3$, since if $\boldsymbol{\theta}^{1:L}$ is not high-regret (which happens with probability $1 - \alpha$), with probability at least $1/3$ $\mathcal{M}(\boldsymbol{\theta}^{1:L})$ will not equal $\text{DB}(\boldsymbol{\theta}^{1:L})$ and thus generate a sequence lying outside the support of $\mathcal{M}_0$. It follows that $d_{TV}(D_0, D) + \mathbb{E}_{\boldsymbol{\theta} \sim D_1}[\text{ExtReg}(\boldsymbol{\pi}, \boldsymbol{\theta})] \geq 1/12$, as desired.

$\square$

## B.6 Proof of Theorem 4.2

*Proof of Theorem 4.2.* First, we bound the TV distance between $D$ and $D_0$. Following the proof of Theorem 3.1, for any substring of length $m \leq \Delta$, the probability that Algorithm 1 plays the out-of-distribution prediction is bounded by $\frac{1}{T} \cdot \frac{1}{\Delta^{\alpha+1}}$. There are at most $\Delta T$ substrings of length bounded by $\Delta$. Applying a union bound we prove the TV distance result.

The external regret bound follows from the same proof in Schneider & Vodrahalli (2024). We write the proof here. Given a context of length $L$, the output of Algorithm 2 can be viewed as the uniform

---

[4]If we want to ensure $D_0$ and $D_1$ have full support, we can add infinitesimal mass on the other option (i.e. follow the de Bruijn sequence with probability $1 - \epsilon$, follow the opposite with probability $\epsilon$). This does not affect any of the subsequent logic.

combination of $\Delta$ copies of Algorithm 1, each starting at a time $m = L + 1, \ldots, L'$. Intuitively, Algorithm 2 inherits the regret of Algorithm 1 over length $\Delta$ strings with the given parameters, which is bounded by $\frac{\sqrt{2}+1}{\Delta} + \sqrt{\frac{8 \log T + 8(\alpha+1) \log \Delta}{\Delta}}$. We denote Algorithm 1 by $\mathcal{A}$ and its regret by $\text{REGRET}_{\mathcal{A}}$. First, we introduce notation related to offsets. For any $t \in -(\Delta - 1), T - 1$ and $m \in [\Delta]$, we write

$$\tilde{a}_t^m = a_{t+m}^m = \mathcal{A}(\theta_t, \ldots, \theta_{t+m-1}),$$

which is the prediction by the $m$-th copy of $\mathcal{A}$ about state $\theta_{t+m}$.

Now we rearrange the total payoff of Algorithm 2 and write it in copies of $\mathcal{A}$:

$$\sum_t \mathbb{I}\left[a_t = \theta_t\right] = \sum_{t=1}^{T} \frac{1}{\Delta} \sum_{m=1}^{\Delta} \mathbb{I}\left[a_t^m = \theta_t\right]$$

$$= \sum_{t=1}^{T} \frac{1}{\Delta} \sum_{m=1}^{\Delta} \mathbb{I}\left[\tilde{a}_{t-m}^m = \theta_t\right]$$

$$= \sum_{t=-\Delta+1}^{T-1} \frac{1}{\Delta} \sum_{m=1}^{\Delta} \mathbb{I}\left[\tilde{a}_t^m = \theta_{t+m}\right]$$

$$\geq \sum_{t=-\Delta+1}^{T-1} \frac{1}{\Delta} \left[\max_{\theta \in \{0,1\}} \sum_{m=t}^{t+\Delta} \mathbb{I}\left[\theta = \theta_m\right] - \Delta \text{REGRET}_{\mathcal{A}}\right]$$

$$\geq \max_{\theta \in \{0,1\}} \sum_{t=1}^{T} \mathbb{I}\left[\theta = \theta_t\right] - (T + M)\text{REGRET}_{\mathcal{A}}.$$

Normalizing both sides by $\frac{1}{T}$, we prove the theorem. $\qquad \square$

## C  TRANSFORMER ROBUSTIFICATION CONSTRUCTION

### C.1  TRANSFORMER PRELIMINARIES

We introduce a formal model of a transformer, drawing heavily from Sanford et al. (2024).

For a sequence of queries, keys, and values $Q, K, V \in \mathbb{R}^{T \times m}$, an *autoregressive self-attention head* of embedding dimension $m$ with softmax attention is defined by

$$f(Q, K, V) = \text{softmax}(QK^{\mathsf{T}})V,$$

where the softmax operator

$$\text{softmax}(v) = \frac{1}{\sum_{i=1}^{T} \exp(v_i)}(\exp(v_1), \ldots, \exp(v_T))$$

is applied row-wise, and mask $M \in \mathbb{R}^{T \times T}$ satisfies

$$M_{i,j} = \begin{cases} 0 & \text{if } i \geq j, \\ -\infty & \text{otherwise.} \end{cases}$$

*Multi-headed attention* concatenates the outputs for multiple attention heads. For some sequential input $X \in \mathbb{R}^{T \times m}$, an $H$-headed attention unit computes $H$ queries, keys, and values of embedding dimension $\frac{m}{H}$ as

$$Q^h = XW_Q^h, \ K^h = XW_K^h, \ V^h = XW_V^h,$$

for projections $W_Q^h, W_K^h, W_V^h \in \mathbb{R}^{m \times m/H}$, for every $h \in [H]$. The output of the resulting $H$-headed attention layer is the following:

$$X \mapsto [f(Q^1, K^1, V^1) \ldots f(Q^H, K^H, V^H)],$$

for parameters $(W_Q^h, W_K^h, W_V^h)_{h \in [H]}$.

We define a *transformer* of depth $L$ as a function of the form

$$g = \phi_L \circ f_L \circ \cdots \circ \phi_1 \circ f_1 \circ \phi_0,$$

where $f_1, \ldots, f_L$ are multi-headed attention layers of embedding dimension $m$, and $\phi_1, \ldots, \phi_{L-1}$ : $\mathbb{R}^m \to \mathbb{R}^m$ are *multi-layer perceptrons* applied element-wise, i.e.

$$\phi_\ell(X) = (\phi_\ell(X_1), \ldots \phi_\ell(X_T)),$$

$\phi_0 : \Sigma \to \mathbb{R}^m$ is an embedding layer from some alphabet $\Sigma$, and $\phi_L : \mathbb{R}^m \to \mathbb{R}^{d_{\text{out}}}$ is an output MLP layer. In the subsequent proof, we argue informally that our MLP units can be efficiently constructed as a shallow ReLU network with bounded width (typically, logarithmic in sequence length $T$) and bit-precision ($\log(T)$ as well).

We assume that the alphabet $\Sigma$ encodes a positional encoding. That is, in the proof of Theorem 5.1, we let $\Sigma = \Theta \times [T]$ and encode the input sequence $\theta^T \in \Theta^T$ as $((\theta_1, 1), \ldots, (\theta_T, T))$. We assume that there exists a constant "beginning-of-sequence token" $X_{\text{BOS}}$ that produces constant key and value vectors and can be attended to.

## C.2 Proof of Theorem 5.1

We restate Theorem 5.1 precisely.

**Theorem C.1.** *Suppose there exists a transformer $g_{\mathcal{M}_0}$ of depth $L$ and embedding dimension $m$ that exactly computes the next-token probabilities over some distribution $D_0$ (i.e. for any $\theta^T \in \Theta^T$, $g_{\mathcal{M}_0}(\theta^T)_{t,i} = \Pr_{D_0}[\theta_t = i \mid \theta^{t-1}]$). Suppose the loss function $U$ is Lipschitz and can be exactly represented by a multi-layer perceptron with width independent of $T$. Then, there exists a transformer $g'$ of depth $L' = L + 4$, heads $H' = O(|A|^2)$, and embedding dimension $m' = m + O(|A|^3 + |\Theta|)$ such that the following is true (in the notation of Algorithm 1) for some error term $\delta \leq \frac{1}{T^c}$ (for any fixed $c > 0$), for all $t \leq T$:*

*1. If there exists $s \leq t - |A|$ such that*

$$\text{REGRET}_s \geq \text{REGRET}_{\text{HEDGE},s} + \frac{1}{\sqrt{T}} \log|A| + \sqrt{8(1+\alpha)(\log T)/s} + \delta,$$

*then $g'(\theta^T)_t = \mathcal{M}_{Polya}(\theta^{(t-1)})$.*

*2. If every $s \leq t - |A|$ satisfies*

$$\text{REGRET}_s < \text{REGRET}_{\text{HEDGE},s} + \frac{1}{\sqrt{T}} \log|A| + \sqrt{8(1+\alpha)(\log T)/s} - \delta,$$

*then $g'(\theta^T)_t = \mathcal{M}_0(\theta^{t-1}) = g_{\mathcal{M}_0}(\theta^T)$.*

Before proving Theorem C.1, we observe that there are two senses in which the transformer construction is approximate:

- The $\text{REGRET}_s$ condition makes no guarantees within an additive interval of width $2\delta$.
- The transformer guarantee does not account for the previous $|A|$ states in the outcomes.

Both issues are insignificant in the regime where $T$ is large, and Theorem 3.1 could be adapted in a straightforward manner to accommodate these changed conditions.

*Proof.* We transformer $g'$ by introducing six gadgets. Assume that $A = \{1, \ldots, k\}$ throughout.

1. The first $L$ layers of $g'$ exactly compute the output of $g_{\mathcal{M}_0}$. Concretely, we assume that the $t$th output of the $L$th layer exactly encodes a positional embedding $u_t$, the input state $\theta_{t-1}$, and the next token distribution under $D_0$:

$$p^t = \left( \Pr_{D_0}[\theta_t = \theta \mid \theta^{t-1}] \right)_{\theta \in \Theta} \in \mathbb{R}^{|\Theta|}.$$

The output MLP computes the expected loss of each action with respect to $p^t$:

$$\ell_a^t = \mathbb{E}_{\theta \sim p^t}[U(a, \theta)], \text{ for each } a \in A.$$

2. An additional head in layer $L$ computes

$$\mathcal{M}_{\text{Polya}}(\boldsymbol{\theta}^{(t-1)}) = \left( \frac{1 + \sum_{s=1}^{t-1} \mathbb{I}\left[ \theta_s = \theta \right]}{|\Theta| + (t-1)} \right)_{\theta \in \Theta}$$

in the $t$th output by calculating a rolling average with the self-attention head. The output MLP computes

$$\ell_a^{\text{HEDGE},t} = \mathbb{E}_{\theta \sim \mathcal{M}_{\text{Polya}}(\boldsymbol{\theta}^{(t-1)})}[U(a, \theta)], \text{ for each } a \in A.$$

3. $k - 1$ heads in layer $L + 1$ jointly retrieve the pairs of partial losses

$$(\ell_1^{t-k+1}, \ell_1^{\text{HEDGE},t-k+1}), (\ell_2^{t-k+2}, \ell_2^{\text{HEDGE},t-k+2}), \ldots, (\ell_{k-1}^{t-1}, \ell_{k-1}^{\text{HEDGE},t-1})$$

from the $k$ previous tokens.

4. Layer $L + 2$ uses $k^2$ attention heads to compute each component of both QBRs.

$$\pi_a^{t-k} = \frac{\exp(\sqrt{T}\ell_a^{t-k})}{\sum_{a'} \exp(\sqrt{T}\ell_{a'}^{t-k})}, \quad \pi_a^{\text{HEDGE},t-k} = \frac{\exp(\sqrt{T}\ell_a^{\text{HEDGE},t-k})}{\sum_{a'} \exp(\sqrt{T}\ell_{a'}^{\text{HEDGE},t-k})}.$$

5. Layer $L + 3$ uses one head to compute $\text{REGRET}_{t-k} - \text{REGRET}_{\text{HEDGE},s}$ by averaging the QBR losses, and evaluate whether the inequality condition holds for $t - k$.

6. Layer $L + 4$ detects whether the inequality condition occurs for *any* $s \le t - k$ by computing an OR over the inequality conditions.

While the proof does not formally define all weights in the model, we outline how each gadget is constructed in the following sections. We focus in greatest specificity on the attention patterns that construct the aggregations employed by different gadgets. We also provide brief justifications for why all MLPs can be compactly constructed and a high-level error analysis.

**Gadget 1: Next-token probabilities (Layers 1 to $L$).** The relative sizes of the two models immediately imply that the first $L$ layers of $g'$ can exactly simulate $g_{\mathcal{M}_0}$. The residual connections in $g'$ (and the slight increase in embedding dimension) make it possible for $g'$ to preserve a positional encoding $u_t$ and $\theta_{t-1}$ throughout the $L$ layers, even if the residual stream of $g_{\mathcal{M}_0}$ "forgets" them.

Because $\ell_a^t$ is a linear function of $p^t$, it can be trivially computed with a linear layer of the $L$th layer's MLP $\phi_L$. The MLP additionally computes

$$(\mathbb{I}\left[ \theta_{t-1} = \theta \right])_{\theta \in \Theta} \in \{0, 1\}^k,$$

which employs $k$ distinct ReLU circuits as fixed thresholds.

**Gadget 2: Polya urn average (Layer $L$).** The Polya urn next-state prediction model (equation 1) can be computed exactly for each state $\theta$ by an attention head that averages over the indicators $\mathbb{I}\left[ \theta_s = \theta \right]$ for $s < t$. The bias of the Polya urn predictor is accounted for by attending to the constant-valued BOS token.

A single autoregressive attention head in the $L$th layer computes $\mathcal{M}_{\text{Polya}}(\boldsymbol{\theta}^{(t-1)})$ by attending to previous tokens (including a BOS token) with the following keys, queries, and values, which are either constant-valued or can be obtained using $O(|\Theta|)$ ReLU neurons as thresholds.

$$Q_t = 1, \quad K_t = 0, \quad V_t = (\mathbb{I}\left[ \theta_{t-1} = \theta \right])_{\theta \in \Theta};$$

$$K_{\text{BOS}} = \log(|\Theta| - 1), \quad V_{\text{BOS}} = \frac{1}{|\Theta| - 1}.$$

These choices produce the following self-attention outputs.

$$\text{softmax}(Q_t K^{\mathsf{T}})V = \frac{\exp(Q_t K_{\text{BOS}})V_{\text{BOS}} + \sum_{s \le t} \exp(Q_t K_s)V_s}{\exp(Q_t K_{\text{BOS}}) + \sum_{s \le t} \exp(Q_t K_s)}$$

$$= \frac{(|\Theta| - 1) \cdot \frac{1}{|\Theta|-1} + \sum_{s \le t} \mathbb{I}\left[ \theta_{s-1} = \theta \right]}{(|\Theta| - 1) + t}$$

$$= \mathcal{M}_{\text{Polya}}(\boldsymbol{\theta}^{(t-1)}).$$

As in Gadget 1, the partial losses $\ell_a^{\text{HEDGE},t}$ can be computed in the output MLP.

Note that none of these quantities depend on $\mathcal{M}_0$; hence, concurrent computation in layer $L$ is possible.

**Gadget 3: Retrieving previous losses (Layer $L+1$).** For each $a \in A$, assume that the positional encoding $u_t$ is sufficiently structured to make possible the retrieval of $u_{t-k+a}$ in an MLP layer. This is possible with simple sinusoidal embeddings (see, e.g., the proof of Theorem 6 of Sanford et al. (2023)). We further assume that $\|u_t\| = 1$ and $u_t^\mathsf{T} u_s \leq 1 - \frac{1}{T^c}$ for some constant $c \geq 0$ if $t \neq s$.

The $a$th attention head has the following components, which can be computed in the MLP of the previous layer:

$$Q_t = T^C u_{t-k+a}, \quad K_t = u_t, \quad V_t = (\ell_a^t, \ell_a^{\text{HEDGE},t}),$$

for any $C \geq c+1$. For any constant $c' > 0$, there exists some sufficiently large $C$ such that the first dimension of the $a$th self-attention output approximately equals $\ell_a^{t-k+a}$:

$$\left| \text{softmax}(Q_t^\mathsf{T} K) V_{\cdot,1} - \ell_a^{t-k+a} \right|$$

$$= \left| \frac{\sum_{s \leq t} \exp(T^C u_{t-k+a}^\mathsf{T} u_s) \ell_a^s}{\sum_{s \leq t} \exp(T^C u_{t-k+a}^\mathsf{T} u_s)} - \ell_a^{t-k+a} \right|$$

$$\leq \left| \frac{\exp(T^C)}{\exp(T^C) + (t-1)\exp(T^C - T^{C-c})} \ell_a^{t-k+a} - \ell_a^{t-k+a} \right| + \left| \frac{(t-1)\exp(T^C - T^{C-c})}{\exp(T^C)} \right|$$

$$\leq \frac{1}{T^{c'}}.$$

We refer back to this inverse-polynomial additive error later when bounding $\delta$. Note that the outputs of this self-attention unit can be computed with bit precision $O(\log T)$.

The analogous claim holds for $v_{\cdot,2}$ and $\ell_i^{\text{HEDGE},t-k+a}$.

**Gadget 4: Computing QBR (Layer $L+2$).** Before formally constructing the QBR predictor $\pi$, we outline how we wish to obtain some $\pi_a^{t-k}$ for some action $a \in A$ in the $t$th sequential position *for a single fixed index $t$* by providing a partial softmax over a subset of $k$ embeddings.

$$\tilde{Q}_t = \sqrt{T}, \quad \tilde{K}_{t-k+a'} = \ell_{a'}^{t-k}, \quad \tilde{V}_{t-k+a'} = \mathbb{I}\left[a' = a\right], \quad \text{for } a' \in A.$$

Note that the previous gadget ensures that sequence element $t - k + a'$ has access to partial loss $\ell_{a'}^{t-k}$. The corresponding softmax exactly computes $\pi_a^{t-k}$.

$$\text{softmax}(\tilde{Q}_t \tilde{K}) \tilde{V} = \frac{\sum_{s=t-k+1}^t \exp(\tilde{Q}_t \tilde{K}_s) \tilde{V}_s}{\sum_{s=t-k+1}^t \exp(\tilde{Q}_t \tilde{K}_s)}$$

$$= \frac{\sum_{a'=1}^k \exp(\tilde{Q}_t \tilde{K}_{t-k+a'}) \tilde{V}_{t-k+a'}}{\sum_{a'=1}^k \exp(\tilde{Q}_t \tilde{K}_{t-k+a'})}$$

$$= \frac{\exp(\sqrt{T} \ell_a^{t-k})}{\sum_{a'=1}^k \exp(\sqrt{T} \ell_{a'}^{t-k})} = \pi_a^{t-k}.$$

This construction in its current is not sufficient because its parameterization depends on a single sequence index $t$, and it attends to only a subset of elements. Two modifications suffice to adapt this construction to compute all sequential outputs.

1. We employ a *width-$k$ interval positional encoding* that the $t$th sequence element only non-negligibly attends to the $k$ previous elements.

2. We use $k^2$ heads such that the $t$th output of the head indexed by $(a, j)$ is $\pi_a^{t-k}$ if $t \equiv j \pmod{k}$ and 0 otherwise.

We assume that the positional encoding $u_t$ can be used to derive a width-$k$ interval encoding $w_t$ that satisfies the following property:

$$w_t^\mathsf{T} u_s = 1 \text{ if } t - k \leq s < t, \text{ and } w_t^\mathsf{T} u_s \leq \frac{1}{2} \text{ otherwise.}$$

These embedding vectors are known to exist and have dimension $O(k)$ by a restricted-isometry condition established by Mendelson et al. (2007); Candes & Tao (2005)[5].

Fix some pair $(a, j) \in [k]^2$. We construct the queries, keys, and values of the corresponding head as follows. We define $a'_{j,t} \in [k]$ as $a'_{j,t} \equiv t - j \pmod{k}$.

$$Q_t = \sqrt{T} w_t, \quad K_t = u_t \left( \ell_{a'_{j,t}}^{t - a'_{j,t}} + c\sqrt{T} \right) - cT, \quad V_t = \mathbb{I}\left[ a'_{j,t} = a \right].$$

Note that the new query, key, and value embeddings are defined for all $t$ and that $V_{t-k+a} = \tilde{V}_{t-k+a}$. Furthermore, the query/key inner-products are preserved within the $k$-interval, and inner-products outside the interval are much smaller under the assumption that $U$ is bounded indepently of $T$. For a sufficiently large constant $c$:

$$Q_t^\mathsf{T} K_s = \sqrt{T} \left( \ell_{a'_{j,t}}^{t - a'_{j,t}} + c\sqrt{T} \right) - cT = \sqrt{T} \ell_{a'_{j,t}}^{t - a'_{j,t}} = \tilde{Q}_t \tilde{K}_s, \qquad \text{if } t - k \leq s < t.$$

$$Q_t^\mathsf{T} K_s \leq \frac{\sqrt{T}}{2} \left( \ell_{a'_{j,t}}^{t - a'_{j,t}} + c\sqrt{T} \right) - cT = \frac{\sqrt{T}}{2} \ell_{a'_{j,t}}^{t - a'_{j,t}} - \frac{cT}{2} \leq -\frac{cT}{4}, \qquad \text{otherwise.}$$

A judicious choice of $c$ ensures that the additive error in the self-attention unit from inner products outside the interval of width $k$ is inversely polynomial in $T$. We conclude the following:

$$\mathrm{softmax}(Q_t^\mathsf{T} K)V = \frac{\sum_{s=1}^{t} \exp(Q_t^\mathsf{T} K_s) V_s}{\sum_{s=1}^{t} \exp(Q_t^\mathsf{T} K_s)} \approx \frac{\sum_{s=t-k}^{t-1} \exp(\tilde{Q}_t^\mathsf{T} \tilde{K}_s) \tilde{V}_s}{\sum_{s=t-k}^{t-1} \exp(\tilde{Q}_t^\mathsf{T} \tilde{K}_s)} = \pi_a^{t-k},$$

where the approximation conceals an additive inverse polynomial error whose degree depends on the choice of $c$, which can be bounded with a similar softmax analysis used in the previous gadget.

Given the QBR distribution $\pi^{t-k}$, the layer's MLP computes $\mathbb{E}_{a \sim \pi^{t-k}}[U(a, \theta_{t-k})]$ by evaluating $\mathbb{E}_{a \sim \pi^{t-k}}[U(a, \theta)]$ for every $\theta \in \Theta$ as a linear function of $\pi^{t-k}$, and using $|\Theta|$ ReLU thresholds to retrieve the correct expectation for $\theta_{t-k}$[6].

Layer $L + 2$ consists of two copies of this gadget. The other one computes $\pi^{\textsc{Hedge}, t-k}$ and

$$\mathbb{E}_{a \sim \pi^{\textsc{Hedge}, t-k}}[U(a, \theta_{t-k})]$$

with $k^2$ additional attention heads and corresponding MLP weights.

**Gadget 5: Evaluating $\textsc{Regret}_{t-k}$ condition (Layer $L + 3$).** Obtaining $\textsc{Regret}_{t-k}$ requires first computing

$$\frac{1}{t - k} \sum_{s \leq t-k} \mathbb{E}_{a \sim \pi^s}[U(a, \theta_s)],$$

which can be attained a transformer that computes a rolling average among $t - k$ preceding elements. The following queries, keys, and values enable that construction:

$$Q_t = 1, \quad K_t = \begin{cases} T & t > k, \\ 0 & t \leq k, \end{cases} \quad V_t = \mathbb{E}_{a \sim \pi^{t-k}}[U(a, \theta_{t-k})].$$

This computes the above quantity up to additive inverse polynomial error.

An analogous computation retrieves the corresponding term for $\textsc{Hedge}$:

$$\frac{1}{t - k} \sum_{s \leq t-k} \mathbb{E}_{a \sim \pi^{\textsc{Hedge}, s}}[U(a, \theta_s)].$$

---

[5]This connection is discussed in detail in Sanford et al. (2023).

[6]This relies on $\theta_{t-k}$ being retrieved from index $t - k + 1$, which is possible with an additional "look-up" attention head that applies the construction of Gadget 3.

The difference in regrets can be computed in the MLP by subtracting the two quantities and scaling appropriately. We conclude by determining whether $\text{REGRET}_{t-k}$ meets the condition. Since the regret is only used as a threshold, we design an MLP that evaluates the following condition:

$$q_{t-k} = \mathbb{I}\left[\text{REGRET}_{t-k} - \text{REGRET}_{\text{HEDGE},t-k} \geq \frac{1}{\sqrt{T}}\log|A| + \sqrt{\frac{8(1+\alpha)(\log T)}{t-k}}\right].$$

Note that the total additive error of the thresholded quantity in the condition is at most inverse polynomial.

**Gadget 6: Determining whether the condition holds anywhere (Layer $L+4$)** The final layer tests whether $q_{s-k} = 1$ for any $s \leq t$ and returns the appropriate distribution based on the result. We employ a single self-attention head with the following components:

$$Q_t = 1, \quad K_t = 2T \cdot q_{t-k}, \quad V_t = 1, \quad k_{\text{BOS}} = T, \quad v_{\text{BOS}} = 0.$$

We set $q_{t-k} = 0$ for $t \leq k$. Consequently, $\text{softmax}(Q_t K^\mathsf{T})V > \frac{2}{3}$ if there exists some $q_{s-k} = 1$ for $s \leq t$, and $\text{softmax}(q_t k^\mathsf{T})v < \frac{1}{3}$ otherwise. Thresholding on this value is sufficient to ensure the that proof claim holds. $\square$

