# OpenReview forum: "Next-Token Prediction and Regret Minimization"
_ICLR.cc/2026/Conference — Submitted to ICLR 2026_

### Official Review · Reviewer_gunY · 2025-10-27

**Soundness:** 3
**Presentation:** 3
**Contribution:** 2
**Rating:** 4
**Confidence:** 3

**Summary:**

This paper studies when and how a model trained for next‑token prediction can be used to make adversarially robust online decisions. The core idea is to define low‑regret distributions, in the sense that the next‑token models whose quantal best‑response induces sublinear adversarial regret when used as a policy. Authors show that (1) the Polya‑urn next‑token model becomes an exponential‑weights learner via quantal best response and thus achieves $1/\sqrt{T}$ average regret bound; (2) a robustification procedure that, given any unbounded‑context model $M_0$ yields a low‑regret model $M$, both of which are close w.r.t. TV distance; (3) For bounded context $L$, robustification can be impossible but becomes feasible if the model can use a slightly longer window. (4) a transformer that solves next‑token prediction can be extended by a few layers to implement the robustified policy.

**Strengths:**

- The motivation of the studied setting looks interesting to me, and this paper asks a natural and practical question: when does next‑token prediction suffice for adversarial online decision making.

- The robustification used in this paper is standard and easy-to-understand. I appriciate authors for explaning intuitions and motivations behind analysis and algorithms, which make this paper easy-to-follow.

- Authors provide experiments to demonstrate that a tiny decoder can learn the robustified behavior and achieve vanishing regret on the stationary and drifting processes.

**Weaknesses:**

- The technical contributions of this paper seem limited to me. For example, making argmax to softmax allows the learner to achieve a low-regret in adversarial setting is quite standard and well-known, while it is stated in the next-token prediction context. Most analysis directly follow previous work or standard ones.

- The learner is too powerful in the studied model, which significantly simplies the problem. In this paper, the learner is assumed to know the utility function and the state chosen by the adversary is also revealed at the end of round. In this case, the learner can even calculate the external regret by herself, which thus enables Algorithm 1 to compare regret bounds. However, assuming the knowledge of these is strong in real-world scenarios since utility function should be learned during interaction.

**Questions:**

- I am curious what if the learner does not know the utility function in advance, but only has a bandit feedback. Do your algorithms still work? If not, what's the main difficulty?

---

> ### Author Response · Authors · 2025-11-20
>
> Thank you for your careful review of our paper. We respond to individual comments below.
>
> * Unknown Utility Function / Bandit Feedback: It is a very interesting question how to extend our results to more general strategic settings involving unknown utility functions / partial (bandit) feedback. We think there are two ways to possibly interpret such an extension.
>
>   - One way to extend our results to “unknown” utility functions is to attempt to provide the stronger guarantee of providing predictions of adversary behavior that result in low-regret for any potential downstream agent (who is approximately best-responding to our predictions according to their unknown utility function). This connects closely with the recent literature on omni-prediction and U-calibration, and we strongly suspect the same methods we introduce in this paper can be extended to this objective.
>
>   - Another way to interpret this question is via the partial feedback / bandit model, where we do not see the action the adversary takes, but only the induced loss of the combination of the learner’s action and the adversary’s action. Since in our setting we study next-token prediction models that predict the adversary’s next move, it is unclear how to apply this model to our setting -- in particular, it is unclear how to generalize the idea of an existing distribution (e.g. specified by a transformer) mapping historical actions to a randomized next state.
>
> * “making argmax to softmax allows the learner to achieve a low-regret in adversarial setting is quite standard...”: We agree this is well-known in online learning (and included it in the preliminaries for this reason). We view our main technical contributions as the theorems in Sections 3, 4, and 5.1 (in addition to introducing this model).

---

### Official Review · Reviewer_38CK · 2025-10-31

**Soundness:** 2
**Presentation:** 3
**Contribution:** 2
**Rating:** 2
**Confidence:** 3

**Summary:**

This paper studies the full-information online learning problem, where the action set is finite and the losses are determined by an oblivious adversary. These losses are parameterized by the sequence $(\theta_t)\_t$. The paper's main idea is simulate the adversary by a next-token prediction algorithm that predicts $\theta_t$ given $\theta_1, \theta_2, \dots, \theta_{t-1}$. A key contribution is the new regret upper bound $O(\sqrt{T} \log(TA))$ in Lemma 2.3. Other key contributions concern the effectiveness of learning to predict $\theta_t$ with bounded/unbounded context length and transformers.

**Strengths:**

- The concept of modelling the adversary as a next-token prediction process is interesting.
- While being sloppy in a few places, the paper is generally well-written.
- The proof of Lemma 2.1 is correct. I did not verify other proofs.

**Weaknesses:**

Major:
- *Weak results:* The best upper bound of the proposed method is $O(log(AT)\sqrt{T})$, which is significantly worse than the $O(\sqrt{T\log(A)})$ of simply running Hedge . This is despite the facts that 1) the proposed method has significantly higher computational cost to run to the algorithm for predicting $\theta_t$ and 2) the proposed method only works for an adversary that chooses losses from a finite, discrete set. As such, both the key theoretical result and the practicality are limited.
- *Unclear motivation:* the online learning setting in the paper considers a non-stochastic adversary, and yet the paper attempts to model how that adversary chooses the losses. The key assumption in adversarial online learning is that the learner makes no statistical assumption about the adversary. As such, there should be no ``pattern" to be learned by a transformer (or any next-token prediction algorithm, for that matter)

Minor: The writing, especially some mathematical terms, is sloppy at times.
- Line 71: Theorem 2.3 -> Lemma 2.3
- Line 126: $U(a, theta_t)$ -> $U(a_t, theta_t)$
- Line 157: I don't understand what $\pi_t = BR(...)$ means. The function $BR(...)$ returns an element of the set $A$, while $\pi_t$ is a distribution over $A$. Did you mean a one-hot vector here?
- Line 178: $U(a, \mu)$ is not defined.

**Questions:**

Please address the concerns and questions raised in the Weaknesses section. Additional comments and questions are below.

Comments:
- The paper would be much stronger if it had a matching lower bound. I suspect that the current approach of using NTP to model the adversary will not be able to do much  better than the $O(\sqrt{T}log(T))$.
- Conceptually, for this line of work on the combination of transformers and online learning, I think that using transformers to learn an online algorithm (i.e Hedge) is much more interesting than trying to use transformers to learn a model of the adversary.

Questions:
- The paper's sections on bounded context length seems interesting. I regret that I did not have more time to look more closely into its proofs. But just from the results alone, they don't seem particularly surprising, given that the down-stream task is adversarial online learning that requires the information of *all* previous rounds. In addition, a summary statistics of previous rounds can be efficiently encoded in a sum (just like Hedge). Can the authors comment on the novelty of these results with bounded context length: can they be derived independently outside of online learning context, for an arbitrary problem where the useful information is arbitrary far away from the current step?

---

> ### Author Response · Authors · 2025-11-20
>
> Thank you for your careful review of our paper. We respond to individual points below:
>
> * “The best upper bound of the proposed method is [...] significantly worse than simply running Hedge”: We emphasize that the goal of our paper is to robustify a given transformer so that it achieves a ``**best of both worlds**’’ guarantee: guaranteeing low-regret against any adversary, while also guaranteeing near-optimal utility against an in-distribution adversary. Simply running Hedge does not achieve any in-distribution guarantee.
>
> * “...the down-stream task is adversarial online learning that requires the information of all previous rounds”: Actually, as we cite, previous work on adversarial online learning has shown that with a context window of size L, you can in general guarantee $O(1/\sqrt(L))$ regret per round. This is $O(T/\sqrt(L))$ regret overall, which is still O(sqrt(T)) in regimes where $L = O(T)$ (e.g. L = T/100, the context is 1/100th of the time horizon). As a result, the fact that it is impossible to get these guarantees in our best-of-both-worlds setting is a priori non-obvious.
>
> * “I think that using transformers to learn an online algorithm (i.e., Hedge) is much more interesting than trying to use transformers to learn a model of the adversary.”: One contribution of our paper is the observation that learning the Polya urn distribution with a transformer *directly* implements Hedge. Implementing the robustified learning algorithms of Algorithm 1 (via e.g. a transformer, like we show is possible in Section 5.1) is strictly harder than implementing the Hedge algorithm.
>
> * Line 157: Yes this should be a one-hot vector -- thanks for pointing this out, we will correct this.

---

> > ### Comment · Reviewer_38CK · 2025-11-27
> >
> > Thanks for the clarification. My follow-up comments are below.
> >
> > 1. I get the point about deriving a BOBW bound, but here I am still not convinced about the result. You did not address the issue that your algorithm assumes that the hypothesis set $\Theta$ is discrete and finite. Hedge does not need this assumption.
> >
> > 2. About your statement: "Polya urn distribution with a transformer directly implements Hedge". Related to the point above: does your Polya urn distribution approach really implement Hedge for *continuous* $\Theta$?
> >
> > 3. Still on the statement "Polya urn distribution with a transformer directly implements Hedge": could you point out which part of your paper (e.g. the appendix) proving that the output of your model is mathematically equivalent to the output of Hedge?
> >
> > 4. Under such a strong assumption on $\Theta$, I would expect the existence of either a matching upper bound to Hedge, or a lower bound saying that this is not possible. Does your work provide any of this?

---

> > > ### Author Response · Authors · 2025-12-03
> > >
> > > We believe that there is some confusion about what we refer to when we talk about our algorithm implementing Hedge (and perhaps more generally, the goals of our paper). In full generality, Hedge is an algorithm for the problem of full-information online learning (“online learning with expert advice”) where every round the learner chooses a mixed action (supported on K actions) and then observes their counterfactual loss for each of those actions.
> > >
> > > But in many important applications of Hedge, the adversary has a finite number of losses to choose from every round (e.g., in repeated games, the adversary induces one loss for each of their possible actions, and in online prediction, the outcome is often assumed to be discrete). We study these online decision making problems in this paper, since they correspond most directly with next-token prediction questions (where it is also generally assumed that there is a finite number of possible tokens). The fact that we can simulate what (some variant) of Hedge would do by noisily best responding to a distribution is primarily a tool to attain the later best-of-both-worlds guarantees, not an attempt to recover the guarantees of Hedge with a next-token prediction model.
> > >
> > > For Q3, we show in the proof of Lemma 2.2 in Appendix B.2 that noisy best responses to the Polya-urn distribution implement the exact same distribution over actions as Hedge would with a specific (not necessarily optimal) learning rate.
> > >
> > > For Q4, we would like to emphasize that the main result of our paper is not Lemma 2.2 (which merely shows that noisy best responses to some NTP model are sufficient to get sublinear regret) but rather the results in Sections 3 and 4 (tradeoffs between TV distance and regret minimization).  So, our focus is not on achieving the optimal regret bounds in Lemma 2.2 (the log (|A|T) * sqrt(T) bound). By tuning the noise parameter eta (based on A), it should be possible to get closer to the optimal regret bound of Hedge.

---

### Official Review · Reviewer_UGWM · 2025-11-01

**Soundness:** 3
**Presentation:** 3
**Contribution:** 2
**Rating:** 6
**Confidence:** 3

**Summary:**

This paper explores whether next-token prediction models can be adapted for adversarial online decision-making tasks with low regret. The authors investigate robustification by modifying a trained model to achieve low adversarial regret while staying close to its original distribution. The paper's main finding is a contrast that this is always possible for models with an unbounded context, but generally impossible for models with a bounded context (like standard transformers). The authors then provide a workaround for the bounded case, prove the robust model can be implemented in a transformer, and validate their claims with illustrative experiments.

**Strengths:**

- Conceptually neat bridge from next-token prediction to no-regret decision making with clear, interpretable constructions
- Mix of positive guarantees, a sharp bounded-context impossibility, and a feasible workaround that clarifies trade-offs
- Transformer realizability plus empirical sketches make the theory more concrete and suggest a practical pathway

**Weaknesses:**

- Empirical validation is toy-level. The binary prediction game with Ber(1/3) to Ber(2/3) switch demonstrates the idea but doesn’t establish practicality for large-scale LLMs. Stronger baselines and realistic tasks would help.
- The bounded-context workaround is somewhat unsatisfactory. Algorithm 2 gets $O(1/\sqrt{\Delta})$ regret by expanding context to $L+\Delta$. This suggests the extra $\Delta$ tokens act as memory for a standard bounded-memory regret algorithm; the robustification of the original bounded model feels less integrated than in the unbounded case.
- Tension between theoretical TV-closeness and practical training. The goal is an exponentially TV-close(D), but training on D must produce behavior that differs on adversarial sequences. The experiments report small Next-Token TV (not exponential). Please clarify how theoretical indistinguishability maps to what’s actually enforced and measured in training.

**Questions:**

Please address the concerns raised above.

---

> ### Author Response · Authors · 2025-11-20
>
> We thank the reviewer for their thoughtful and careful review. We respond to individual concerns/weaknesses below.
>
> * Weakness 1: We think the question of understanding whether these empirical results extend to larger-scale transformers and language models is an excellent question for future work. Our goal in this paper is primarily to propose a theoretical model for robustifying next-token prediction models in strategic settings, and our experiments serve primarily to complement these results (see also the response to W3 below).
>
>
> * Weakness 2: The extra tokens notably do *not* act as additional memory for a standard bounded-memory regret algorithm. Indeed, central to our analysis of the bounded context case is the assumption that the learner’s action can only depend on the *opponent’s* previous L actions (and not additional memory the learner can store/modify across time). So while it is true that an online learning algorithm can obtain low regret while just maintaining a low-memory summary of the cumulative utility of each action, this is distinct from what the algorithms we implement accomplish.
>
>
> * Weakness 3: We would like to emphasize that the purpose of our experiments in Section 5 is not to directly reproduce the guarantees one would obtain from directly implementing the robustification in Algorithm 1. Instead (as we mention at the beginning of Section 5), it is an empirical investigation into whether it is possible to *train* transformer models with similar properties -- after all, in practice we don’t want to be running an online learning algorithm like Hedge in parallel to a language model, but rather we want the robust properties of Hedge to be learned by the model itself. Our theoretical work in Section 5.1 and experiments in Section 5.2 show that this is indeed possible in simple settings.

---

### Official Review · Reviewer_1pVk · 2025-11-01

**Soundness:** 3
**Presentation:** 3
**Contribution:** 2
**Rating:** 6
**Confidence:** 3

**Summary:**

The paper studies training an LLM-style next-token predictor on sequences of opponent actions, when does predict to best respond give adversarial no-regret? It shows (1) there exist next-token models  such that a quantal best response to the model exactly simulates Hedge and gets low regret. (2) In the unbounded-context case, every model can be robustified to produces a new model that is low-regret while staying TV-close to the original model. (3) In the bounded-context case with size $L$, this robustification is in general impossible. (4) A transformer can implement the unbounded robustifier with $L+4$ layers and small width increase.

**Strengths:**

1. The paper makes formulate a new question “Is next-token prediction enough to get no-regret?” It gives analysis showing that it is possible but only if we can modify the model on rare bad prefixes. The Polya-urn to QBR to Hedge reduction serves as the key technique for this problem.

2. Robustification theorem is tight. In unbounded context we can turn any model into a low-regret one when the prediction quality is essentially preserved.

3. The authors also provide impossibility for bounded context. Some $L$-bounded models can’t be made low-regret without leaving the class. This gives a nontrivial separation.

4. The authors also show that we you can implement the robustifier with a standard transformer (+4 layers) makes the theory relevant to practice, and the experiments give some evidence to this idea.

**Weaknesses:**

1. The paper studies full information setting where after each interaction, the state $\theta_t$ chosen by the adversary is revealed to the learner. It is unclear for the more challenging bandit feedback case.

2. The detect high regret and switch to Polya logic assumes access to the entire past to compute both the main model’s regret and the Polya/Hedge benchmark. That’s why it works in unbounded context. This is exactly what breaks in realistic LLMs that cap context. The paper proves this limitation, but it also means the positive result hinges on an assumption modern models routinely violate.

3. The main claim ensure TV-close but the experiments themselves show that even two transformers trained on the same Bernoulli process can have large sequence-level TV, only next-token TV is small. That suggests the formal closeness guarantee may be much stricter than what current training can realize.

4. The overall technical contribution is limited. It combines existing online-learning ideas, not in a new learning algorithm.

**Questions:**

1.  Lemma 2.2 picks $U$ after seeing the model. Do your positive robustification results still hold if $U$ is fixed before seeing the model, or do we need to know $U$ to build the switch rule?

2. Since full-sequence TV is very unforgiving, is it better to restate Theorem 3.1 in terms of expected next-token TV (your Table 2 metric) and prove that’s what’s actually preserved by Algorithm 1?

---

> ### Author Response · Authors · 2025-11-20
>
> We thank the reviewer for their thoughtful and careful review. We respond to individual questions below.
>
> * Q1 (picking U in Lemma 2.2): The function U does not need to depend on the model M (in the current proof, U is picked to be the “Adversarial Online Prediction” loss $U(a, \theta) = 1[a = \theta]$ independently of M). We will adjust the statement of Lemma 2.2 to make this clear.
>
> * Q2 (theoretical guarantee on next-token TV): One thing that we should make clearer is that the “next-token TV distance” in the experimental section is a *weaker* metric than the full TV distance. In particular, our (exponentially small) TV distance guarantee in Theorem 3.1 implies an identical guarantee for next-token TV distance. We measure the weaker next-token TV distance in our experiments in Section 5 because, in general, we cannot expect our trained models to attain the same guarantee as the idealized information-theoretic guarantees of Theorem 3.1.

---

### Comment · Area_Chair_j1cp · 2025-11-27
**Reminder: Please Discuss**

All Reviewers,

Thank you for your time. As the rebuttal has been available for a while, please engage in discussions with the authors and with one another. There are only a few days left before December 3.

Best,
Area Chair

---

### Author Response · Authors · 2025-12-03
**Summary for AC**

We thank the AC for the service, especially at this time. We responded to all reviewers' questions. In summary, our paper shows how next-token prediction can achieve the Best-of-Both-Worlds guarantee: the prediction guarantees no regret in an adversarial world, and still remains close to the in-distribution prediction in the stochastic world. Our empirical simulation validates our theoretical findings.

We would like to highlight the following discussions regarding the discussions with reviewers.

* Generality of the results (1pVk, 38CK, gunY).
  - (Unknown utility function) Reviewer gunY asks whether the results extend to the case where the utility function is unknown. We think our result naturally extends to providing no-regret guarantees for every possible downstream decision maker. We will add it in a future version of our paper.
  - (Comparison to multi-armed bandit) Reviewer 1pVk asks about the generalization to partial feedback under the multi-armed bandit problem; 38CK asks about the comparison with the bound in the multi-armed bandit problem.

    We highlight the distinction between the two settings. We are motivated by the prediction problem with an autoregressive model. The autoregressive model predicts a sequence of states of the world, or the action of an adversary. This problem naturally fits under the setting of online prediction, where the goal is to predict a state from a finite set under a fixed loss function. It is unclear to us how the bandit setting applies to our prediction problem, where the goal of multi-armed bandits with full feedback is to select an action under arbitrary losses in each round.
* Optimality of the results (38CK). The reviewer asks about the additional log T factor in the regret bound, compared to the guarantee of Hedge for the online prediction problem. We clarify that our paper does not simply focus on obtaining regret bounds. Our goal is to achieve the best-of-both-worlds guarantee both in distribution and adversarially. Aside from the regret bound, our main result is 1) a construction where the simple, learnable Polya urn distribution implements a no-regret algorithm, and 2) a TV-distance bound of the predicted distribution from the in-distribution predictions. We note that the additional log T factor comes from the reduction in 1). Also, simply running Hedge or optimizing for no-regret does not guarantee 2).

---

### Meta-Review · Area_Chair_qWU9 · 2026-01-16

**Summary:**

The paper looks at adversarial online next token prediction problem for transformers (auto-regressive models). The model is trained on some target distribution over sequences, but the goal is to also have low regret against adversarial sequences -- a best of both worlds kind of approach. There could have been more motivation of this problem and discussion. There are theoretical results that shows that with arbitrary context-window length non-trivial regret minimization can be achieved. However, this is not in general possible when the predictions are based on only limited context window (though slightly elongating the context window gives some guarantees).

**Reviewer Concerns:**

Reviewers raised concerns regarding motivation, comparisons to existing worst-case algorithms and empirical evaluation.

**Reviewer Scores:**

The scores would have been slightly revised upwards, but not sufficiently to merit clear acceptance.

---

### Decision · Program_Chairs · 2026-01-26

Reject